

# Dynamic decay and superadiabatic forces in the van Hove dynamics of bulk hard sphere fluids

**Lucas L. Treffenstädt[1], Thomas Schindler[2] and Matthias Schmidt[1]⋆**

**1** Theoretische Physik II, Universität Bayreuth, Universitätsstr. 30, 95447 Bayreuth, Germany
**2** Theoretische Physik I, FAU Erlangen-Nürnberg, 91058 Erlangen, Germany

⋆ Matthias.Schmidt@uni-bayreuth.de

## Abstract

We study the dynamical decay of the van Hove function of Brownian hard spheres using event-driven Brownian dynamics simulations and dynamic test particle theory. Relevant decays mechanisms include deconfinement of the self particle, decay of correlation shells, and shell drift. Comparison to results for the Lennard-Jones system indicates the generality of these mechanisms for dense overdamped liquids. We use dynamical density functional theory on the basis of the Rosenfeld functional with self interaction correction. Superadiabatic forces are analysed using a recent power functional approximation. The power functional yields a modified Einstein long-time self diffusion coefficient in good agreement with simulation data.



# 1 Introduction

Even a homogeneous liquid at equilibrium has microscopic motion. Although the average one-body density profile is constant in space and in time, the underlying particle motion is vigorous at the particle scale. The van Hove function is the fundamental two-body correlation function to characterise the dynamics of bulk liquids [1–3]. Given a particle at the origin at time $t = 0$, the van Hove function $G(r, t)$ gives the probability density of finding a particle at a distance $r$ away from the origin at time $t$. The van Hove function can be measured experimentally via confocal microscopy [2,4] or by measuring its Fourier transform, the intermediate scattering function, and then inverse Fourier transforming to real space [3,5]. Studying the van Hove function yields significant insight into the dynamics of simple and complex systems. Notable examples thereof include cage formation in nematic and smectic liquid crystals [6], de Gennes narrowing of liquid iron [7], self-motion of water [5] and the dynamics of colloid-polymer mixtures [8]. Likewise, much effort has been made to gain a theoretical understanding of the dynamics of the van Hove function itself. Medina-Noyola and coworkers presented a series of insightful studies based on generalised Langevin equations [9–18]. Weysser *et al.* used mode-coupling theory in comparison to Langevin dynamics simulations [19]. The closely related problem of complex memory in molecular dynamics has recently received much attention [20–24].

For $t = 0$, the van Hove function is proportional to the bulk fluid radial distribution function $g(r) \propto G(r, 0)$. The asymptotic decay of $g(r)$ at large distances $r$ has been the subject of intensive study for a broad range of model liquids. Fisher and Widom, when studying in a one-dimensional model the decay of correlations at walls, found that a line in the temperature-density plane separates the phase diagram into a region of monotonic decay and a region of oscillatory decay [25]. The Fisher-Widom line separates the two classes of universal decay, in the sense that the type of decay applies not only to the behaviour of liquids at interfaces, but also to the large distance behaviour of the bulk liquid radial distribution function [26,27], see e.g. [28–32] for studies in a variety of systems.

Classical density functional theory (DFT) [33,34] has proven to be a powerful tool in the study of soft condensed matter [35]. A particular success was the development of fundamental measure theory (FMT) for the description of the hard sphere liquid [35–38]. DFT provides two pathways to calculating the radial distribution function. The Ornstein-Zernike equation relates the radial distribution function to the direct correlation function [3], which can be calculated from functionally differentiating the free energy functional [3]. This path was recently shown to be accessible even in inhomogeneous situations [39,40]. (A nonequilibrium Ornstein-Zernike equation for time-dependent systems was derived by Brader and Schmidt [41,42].) Alternatively to the Ornstein-Zernike route, the radial distribution function can be calculated via Percus' static test particle limit [43]. Here, a system is put under the influence of an external potential that corresponds to the pair potential of one particle

fixed at the origin. The equilibrium density distribution of the system is then proportional to $g(r)$ [3, 43]. For applications of the static test part limit, see e.g. [28, 29, 44, 45].

A similar mapping of two-body correlations to one-body distributions is possible for the van Hove function. This dynamic test particle limit was first presented using DDFT [46, 47]. DDFT [34, 48, 49] is an extension of DFT, which is based upon approximating the effects of interparticle interactions as those calculated from a free energy functional via the adiabatic construction [50], in order to obtain the time evolution of a non-equilibrium system. The dynamic test particle limit, just as its static counterpart, splits all particles in a system into a single tagged ('self') particle and the remaining ('distinct') particles. Starting from an equilibrium configuration with the self particle fixed at the origin, the self particle is released and the system evolves in time. Its one-body density profile is then equal to the van Hove function. The dynamic test particle approach was implemented for the Lennard-Jones liquid within Brownian dynamics simulations by Schindler and Schmidt [51]. Brader and Schmidt presented a formally exact formulation of the dynamic test particle limit using power functional theory (PFT) [52, 53]. The power functional framework provides a systematic way to improve upon the well-known defects of DDFT, such as the overestimation of relaxation rates [48, 54]. At the core of PFT is the formally exact splitting of the time-dependent one-body internal force field into adiabatic and superadiabatic contributions. The former can be obtained from the equilibrium free energy functional, and the latter constitute genuine nonequilibrium, flow-dependent forces that are generated by the superadiabatic power functional.

There is much recent and renewed interest in the dynamics of the van Hove function. Stopper *et al.* studied the van Hove function of a hard sphere liquid at densities up to $0.76\sigma^{-3}$, using dynamic density functional theory (DDFT) with a partially linearised White Bear Mk. 2 excess free energy functional [55]. The authors also carried out kinetic Monte Carlo simulations. They subsequently improved upon their approach by introducing a 'quenched' excess free energy functional to address the problem of self interactions within the self density component. Furthermore, they introduced an inhomogeneous particle mobility correction to the DDFT equation of motion [56]. More recently, they extended their approach to two-dimensional hard discs and found good agreement of their results with experimental data obtained by video microscopy of a two-dimensional colloidal suspension of melamine formaldehyde particles [4].

Despite these successes, studies of the van Hove function with DDFT systematically neglect superadiabatic effects by construction [46, 47]. Recently, we have combined viscoelastic, drag-like and structural forces to obtain an approximation for superadiabatic forces in the dynamics of the hard sphere van Hove function [57]. This work builds upon an approximation developed for viscoelastic forces in a sheared system of hard spheres [58] and approximations developed for drag-like and structural forces in a binary system of counter-driven hard spheres [59]. In [57], we have shown that superadiabatic forces contribute significantly to the internal force field at all times. Our approximation describes the internal force field accurately starting at short times $t > 0.3\tau$ up to at least $t = 1.5\tau$, where $\tau \equiv \sigma^2/D$ is the intrinsic Brownian timescale of the system, with particle diameter $\sigma$ and diffusion constant $D$.

Here, we expand on our previous study [57]. We investigate the van Hove function of a bulk hard sphere liquid at bulk density $\rho_B \approx 0.73\sigma^{-3}$ (packing fraction $\eta \approx 0.38$). Although the hard sphere model liquid is simple, both its static and dynamic properties are often considered to be universal and taken to represent a much wider range of systems [3, 60]. Using event-driven Brownian dynamics simulations (BD) [61], we obtain the van Hove function itself as well as the internal force field acting in this system. We compare the dynamic structural decay of the van Hove function for hard spheres with results for the Lennard Jones liquid [51]. The adiabatic contributions to the internal force field are calculated using Monte Carlo simulations [62, 63] and DDFT [48, 49] in the dynamic test particle limit. For the latter method, we evaluate the accuracy of two modifications of the free energy functional, which correct for

unphysical self interactions of the test particle within the dynamic test particle limit [55, 56]. Since adiabatic forces vanish in the long-time behaviour of the van Hove function, the DDFT approximation asymptotically approaches ideal diffusion, and hence fails to model collective slowing down. Therefore, we examine the superadiabatic contributions to the internal force field that governs the time evolution of the van Hove function. The construction of our previously presented PFT approximation [57] for the superadiabatic force field is shown in detail. Finally, we use the PFT approximation to calculate the long-time diffusion constant of the self part of the van Hove function and compare it to simulation results to evaluate the accuracy of the approximation.

This paper is structured as follows: Section 2 describes in detail the models, theory and algorithms used in our work. Section 3 presents results. Starting with the radial distribution function, we examine the dynamic structural decay of the hard sphere van Hove function in BD and DDFT and also compare with the van Hove function of the Lennard-Jones liquid. Lastly, we identify adiabatic and superadiabatic contributions to the internal force field using MC simulation and present our PFT approximation for superadiabatic forces. We summarise and give an outlook in section 4.

## 2 Model and Theory

### 2.1 Brownian Dynamics

We consider a system of $N$ monodisperse hard spheres with diameter $\sigma$. The particle positions $\mathbf{r}_1, \ldots, \mathbf{r}_N \equiv \mathbf{r}^N$ evolve in time according to the Langevin equation of motion

$$\gamma \dot{\mathbf{r}}_i(t) = \mathbf{f}_{\text{int},i}(\mathbf{r}^N) + \mathbf{f}_{\text{ext}}(\mathbf{r}_i, t) + \sqrt{2\gamma k_{\text{B}} T} \mathbf{R}_i(t), \tag{1}$$

where $\gamma \equiv k_{\text{B}} T / D$ is the friction constant of the particles against the implicit solvent, $k_B$ is Boltzmann's constant, $T$ is the temperature, and $\mathbf{f}_{\text{int},i}(\mathbf{r}^N) = -\nabla_i u(\mathbf{r}^N)$ is the internal force that all other particles exert on particle $i = 1, \ldots, N$ due to the interparticle interaction potential $u(\mathbf{r}^N)$. Furthermore, $\mathbf{f}_{\text{ext}}(\mathbf{r}, t)$ is an external one-body force field that in general drives the system out of equilibrium and $\mathbf{R}_i(t)$ is a delta-correlated Gaussian white noise with $\langle \mathbf{R}_i(t) \rangle = 0$ and $\langle \mathbf{R}_i(t) \mathbf{R}_j(t') \rangle = \delta(t - t') \delta_{ij} \mathbb{1}$, where $\delta(\cdot)$ is the Dirac distribution, $\delta_{ij}$ indicates the Kronecker delta, and $\mathbb{1}$ is the $3 \times 3$ unit matrix. The intrinsic (Brownian) timescale of the system is $\tau$.

Since for hard spheres the interparticle interaction potential is discontinuous at contact, integration of the equation of motion (1) requires specifically adapted algorithms. One state-of-the-art algorithm is event-driven Brownian dynamics [61], which we apply to the bulk dynamics where $\mathbf{f}_{\text{ext}}(\mathbf{r}, t) = 0$. We choose a fixed timestep $\Delta t$. (See [64] for adaptive Brownian dynamics for soft potentials.) At the beginning of a simulation step, the particle velocities are randomised according to the Maxwell distribution. The particles are then moved according to the laws of ballistic motion with elastic collisions. Potential particle collisions are detected in advance and handled in the order at which they occur. Once the time $\Delta t$ has passed in the simulation timeframe, the particle velocities are again randomised and the process is repeated.

### 2.2 One- and Two-Body Correlation Functions

In a general nonequilibrium situation, the behaviour of the liquid can be characterised by the time-dependent one-body density and current distributions. The density distribution is defined as

$$\rho(\mathbf{r}, t) = \left\langle \sum_{i=1}^{N} \delta(\mathbf{r} - \mathbf{r}_i(t)) \right\rangle, \tag{2}$$

where $\langle \cdot \rangle$ indicates an instantaneous average over the noise and over initial microstates. The one-body current distribution is defined as

$$\mathbf{J}(\mathbf{r}, t) = \left\langle \sum_{i=1}^{N} \delta(\mathbf{r} - \mathbf{r}_i(t)) \mathbf{v}_i(t) \right\rangle, \tag{3}$$

where, in a numerical simulation, the velocity $\mathbf{v}_i(t)$ of particle $i$ must be calculated with a finite difference of the particle position, centred at time $t$ [65]. We calculate the current distribution in our BD simulations in this manner. The density distribution is connected to the current distribution via the continuity equation

$$\frac{\partial}{\partial t} \rho(\mathbf{r}, t) = -\nabla \cdot \mathbf{J}(\mathbf{r}, t), \tag{4}$$

where $\nabla$ denotes the derivative with respect to $\mathbf{r}$. Given a current $\mathbf{J}(\mathbf{r}, t)$ and an initial condition $\rho(\mathbf{r}, 0)$, the time evolution of the density distribution follows from the continuity equation. In this work, we study a liquid in equilibrium without external fields, so the total density field is always constant and the total current distribution vanishes at all times. However, using the dynamic test particle limit [53] enables us to express time-dependent two-body quantities via one-body quantities in a suitably constructed setup, which then acquires an inhomogeneous density distribution and nonzero current (see Sec. 2.3).

The van Hove function [1, 3] is a dynamical two-body correlation function, defined for a bulk fluid as

$$G(\mathbf{r}, t) = \frac{1}{N} \left\langle \sum_{i=1}^{N} \sum_{j=1}^{N} \delta\left(\mathbf{r} + \mathbf{r}_j(0) - \mathbf{r}_i(t)\right) \right\rangle. \tag{5}$$

The van Hove function measures the probability of finding a particle at $\mathbf{r}$ at time $t$, given that there was a particle at the origin at time zero. In equilibrium without external forces, $G(\mathbf{r}, t)$ is radially symmetric and thus depends only on the modulus $r = |\mathbf{r}|$, i.e. $G(r, t)$. The double sum in (5) can be split according to

$$G(\mathbf{r}, t) = \frac{1}{N} \left\langle \sum_{i=1}^{N} \delta\left(\mathbf{r} + \mathbf{r}_i(0) - \mathbf{r}_i(t)\right) \right\rangle + \frac{1}{N} \left\langle \sum_{i=1}^{N} \sum_{j \neq i}^{N} \delta\left(\mathbf{r} + \mathbf{r}_j(0) - \mathbf{r}_i(t)\right) \right\rangle \tag{6}$$

$$\equiv G_{\mathrm{s}}(\mathbf{r}, t) + G_{\mathrm{d}}(\mathbf{r}, t), \tag{7}$$

where $G_{\mathrm{s}}(\mathbf{r}, t)$ is the self part of the van Hove function and $G_{\mathrm{d}}(\mathbf{r}, t)$ is its distinct part.

## 2.3 Dynamic Test Particle Limit

While the van Hove function is a dynamical two-body correlation function, it can be equivalently expressed in terms of time-dependent one-body quantities of a system with a specifically constructed initial condition [46]. In a system of $N$ identical particles in volume $V$, one particular particle is selected as the test (or self) particle. The system is prepared such that the test particle is at the origin, with the $N - 1$ remaining particles (the distinct particles) being in equilibrium around the test particle. For $N, V \to \infty$ with the bulk number density $\rho_B = N/V = \mathrm{const}$, the self particle is distributed according to the self density distribution

$$\rho_{\mathrm{s}}(\mathbf{r}, 0) = \delta(\mathbf{r}), \tag{8}$$

and the distinct particles are distributed according to the distinct density distribution

$$\rho_{\mathrm{d}}(\mathbf{r}, 0) = \rho_{\mathrm{B}} g(r), \tag{9}$$

where the normalisation is such that

$$\int_V d\mathbf{r}\,\rho_s(\mathbf{r},t) = 1\,, \tag{10}$$

$$\int_V d\mathbf{r}\,\rho_d(\mathbf{r},t) = N-1\,. \tag{11}$$

By starting from this special initial condition, we obtain for the self density distribution the correspondence

$$\rho_s(\mathbf{r},t) = G_s(\mathbf{r},t)\,, \tag{12}$$

whereas the distinct particles have the density distribution

$$\rho_d(\mathbf{r},t) = G_d(\mathbf{r},t)\,. \tag{13}$$

The test particle correspondence can be used to calculate the van Hove function with an approach that delivers microscopic dynamics on the one-body level, such as DDFT or PFT (see Secs. 2.4, 2.5). The time evolution can be expressed in terms of self and distinct currents $\mathbf{J}_\alpha(\mathbf{r},t)$, where $\alpha = s,d$ is a label for the self or distinct part. The continuity equation relates van Hove current and density according to

$$\frac{\partial}{\partial t}G_\alpha(\mathbf{r},t) = -\nabla\cdot\mathbf{J}_\alpha(\mathbf{r},t)\,. \tag{14}$$

The current arises from a force balance relationship [53,66]

$$\gamma\mathbf{J}_\alpha(\mathbf{r},t) = -k_B T\nabla G_\alpha(\mathbf{r},t) + G_\alpha(\mathbf{r},t)\mathbf{f}_{int,\alpha}(\mathbf{r},t)\,, \tag{15}$$

where $-k_B T\nabla G_\alpha(\mathbf{r},t)$ is the ideal force density and $\mathbf{f}_{int,\alpha}(\mathbf{r},t)$ is the internal force acting on $G_\alpha(\mathbf{r},t)$. The (species-labelled) internal force density is the product

$$\mathbf{F}_{int,\alpha}(\mathbf{r},t) = G_\alpha(\mathbf{r},t)\mathbf{f}_{int,\alpha}(\mathbf{r},t)\,. \tag{16}$$

In BD simulation, we can sample the species-labelled current using (3) by considering either only the distinct particles or only the self particle. Results for $\mathbf{F}_{int,\alpha}(\mathbf{r},t)$ can then be calculated using (15).

The force density $\mathbf{F}_{int,\alpha}(\mathbf{r},t)$ consists of an adiabatic and a superadiabatic contribution,

$$\mathbf{F}_{int,\alpha}(\mathbf{r},t) = \mathbf{F}_{ad,\alpha}(\mathbf{r},t) + \mathbf{F}_{sup,\alpha}(\mathbf{r},t)\,, \tag{17}$$

where the adiabatic force density $\mathbf{F}_{ad,\alpha}(\mathbf{r},t)$ is defined via the adiabatic construction [50]: At each fixed point in time $t$, one chooses a pair of external potentials $V_{ad,s}(\mathbf{r},t)$ and $V_{ad,d}(\mathbf{r},t)$ that act on the test particle and on the distinct particles respectively, such that the equilibrium densities $\rho_s(\mathbf{r},t)$ and $\rho_d(\mathbf{r},t)$ under the influence of these potentials match the van Hove function at that point in time. Hence the matching condition is

$$\rho_\alpha(\mathbf{r},t) = G_\alpha(\mathbf{r},t)\,. \tag{18}$$

The adiabatic force field $\mathbf{f}_{ad,\alpha}(\mathbf{r},t) \equiv \mathbf{F}_{ad,\alpha}(\mathbf{r},t)/\rho_\alpha(\mathbf{r},t)$ is then defined as the internal force acting in this equilibrium system [50]. This force can be identified from a known adiabatic external potential $V_{ad,\alpha}(\mathbf{r},t)$ via

$$\mathbf{f}_{ad,\alpha}(\mathbf{r},t) = \nabla V_{ad,\alpha}(\mathbf{r},t) + k_B T\nabla\ln\left(\rho_\alpha(\mathbf{r},t)\right)\,. \tag{19}$$

For a known internal force density $\mathbf{F}_{int,\alpha}(\mathbf{r},t)$, the superadiabatic force density can then be easily calculated by subtracting $\mathbf{F}_{ad,\alpha}(\mathbf{r},t)$ from $\mathbf{F}_{int,\alpha}(\mathbf{r},t)$ (see Eq. (17)).

The adiabatic construction can be performed either with DFT or with particle-based computer simulation. In DFT, the adiabatic force field is directly available from the excess free energy functional via

$$\mathbf{f}_{\mathrm{ad},\alpha}(\mathbf{r}, t) = -\nabla \frac{\delta F_{\mathrm{exc}}[\rho_s, \rho_d]}{\delta \rho_\alpha(\mathbf{r}, t)}. \tag{20}$$

In simulation, iterative methods can be applied to find the adiabatic potentials $V_{\mathrm{ad},s}(\mathbf{r}, t)$ and $V_{\mathrm{ad},d}(\mathbf{r}, t)$ [65,67]. Using simulations has the advantage of being exact up to stochastic fluctuations and finite size effects. We choose this procedure to analyse adiabatic force profiles for a few representative points in the time evolution of the van Hove function, using Metropolis Monte Carlo simulation to obtain the adiabatic density profiles.

## 2.4 Dynamic Density Functional Theory

DDFT [34, 48, 49] is a method of calculating the time evolution of a one-body density distribution. In the case of an $M$-component mixture, we have $M$ density components $\rho_\alpha(\mathbf{r}, t)$ and $M$ current components $\mathbf{J}_\alpha(\mathbf{r}, t)$ with $\alpha = 1, \ldots, M$. Each current component is approximated by the adiabatic current

$$\gamma \mathbf{J}_\alpha^{\mathrm{ad}}(\mathbf{r}, t) = \rho_\alpha(\mathbf{r}, t) \left[ \mathbf{f}_{\mathrm{ext},\alpha}(\mathbf{r}, t) - \nabla \frac{\delta F[\{\rho_\alpha(\mathbf{r}, t)\}]}{\delta \rho_\alpha(\mathbf{r}, t)} \right], \tag{21}$$

where $\mathbf{f}_{\mathrm{ext},\alpha}(\mathbf{r}, t)$ is an external force on component $\alpha$ and $F[\{\rho_\alpha\}]$ is the free energy functional

$$F[\{\rho_\alpha\}] = F_{\mathrm{exc}}[\{\rho_\alpha\}] + \sum_{\alpha=1}^{M} k_B T \int d\mathbf{r} \rho_\alpha(\mathbf{r}, t) \left( \ln \Lambda^3 \rho_\alpha(\mathbf{r}, t) - 1 \right), \tag{22}$$

with thermal de Broglie wavelength $\Lambda$ and excess free energy functional $F_{\mathrm{exc}}[\{\rho_\alpha\}]$. By inserting (21) and (22) into (4), we obtain the DDFT equation of motion [34, 48, 49]

$$\gamma \frac{\partial}{\partial t} \rho_\alpha(\mathbf{r}, t) = k_B T \nabla^2 \rho_\alpha(\mathbf{r}, t) + \nabla \cdot \rho_\alpha(\mathbf{r}, t) \left( \nabla \frac{\delta F_{\mathrm{exc}}[\{\rho_\alpha\}]}{\delta \rho_\alpha(\mathbf{r}, t)} - \mathbf{f}_{\mathrm{ext},\alpha}(\mathbf{r}, t) \right). \tag{23}$$

Here, the implicit assumption is made that the adiabatic current is a fair representation of the total current in the system. However, in many cases, this assumption does not hold (see e.g. [50, 58, 68]) and we will investigate this point for the present problem below.

The test particle limit can be treated in DDFT as a special case of a binary mixture. The density is then a mixture of a self component $\rho_s(\mathbf{r}, t)$, representing the test particle, and a distinct component $\rho_d(\mathbf{r}, t)$, representing the rest of the system. The accuracy of results obtained using DDFT hinges upon having a reliable approximation of the excess free energy functional, which is not known exactly for any kind of interacting particles, save for e.g. the case of hard rods in one dimension [3, 38, 69, 70]. Since the interactions of the constituent particles of both of the density components in the test particle limit are identical, one could think that the excess free energy functional is simply $F_{\mathrm{exc}}[\rho_s, \rho_d] = F_{\mathrm{exc}}[\rho_s + \rho_d]$. One would then apply the appropriate one-component excess free energy functional. However, equilibrium DFT is constructed in the grand canonical ensemble, which means that a density profile normalised to unity represents an average of one particle, instead of exactly one particle [71]. This leads to unphysical self-interactions within the self density component [47].

Stopper *et al*. proposed in Ref. [55] a partially linearised approach, where, for the calculation of the forces on the self density component, the contribution of $\rho_s(\mathbf{r}, t)$ to the excess free energy functional is expanded around $\rho_s(\mathbf{r}, t) = 0$, while the force field acting on the distinct component is calculated using the full, i.e. unlinearised, functional. In a follow-up paper [56], they furthermore proposed a 'quenched' free energy functional

$$F_{\mathrm{exc},q}[\rho_s, \rho_d] = F_{\mathrm{exc}}[\rho_s + \rho_d] - F_{\mathrm{exc}}[\rho_s], \tag{24}$$

and found that the accuracy of their results for $G(\mathbf{r}, t)$ were greatly increased over the prior method. To further evaluate the accuracy of both the partially linearised and the quenched free energy functional, we calculate reference data for the adiabatic force field using Monte Carlo simulation. This allows us to compare these quasi-exact results to approximate forces obtained using the free energy functional. As a base one-component functional, we choose the White Bear Mk. 2 free energy functional [72] with tensorial modification [73], which is known to perform very well, even up to high densities, see e.g. [74].

## 2.5 Power Functional Theory

The phenomenological equation of motion (23) of DDFT does not capture the full non-equilibrium dynamics of many-particle systems. Important physical effects such as drag [59, 83, 84], viscosity [58, 75] and structural non-equilibrium forces [59, 76–82] are absent. PFT provides a formally exact method for including such effects and for calculating the full current in a non-equilibrium system [52]; see Ref. [66] for a pedagogical introduction to the framework. Both adiabatic forces, which give rise to the adiabatic current $\mathbf{J}^{\mathrm{ad}}(\mathbf{r}, t)$ via equation (21), but also superadiabatic forces, which characteristically depend functionally on both the density profile *and* on the current distribution, are included. It has been shown that superadiabatic forces can be of very significant magnitude and that they are in general not trivially related to the adiabatic forces [68, 76].

The full current of an overdamped $M$-component liquid can be calculated in PFT in principle from a functional derivative of the total free power functional $R_t[\{\rho_\alpha, \mathbf{J}_\alpha\}]$ via

$$\frac{\delta R_t[\{\rho_\alpha, \mathbf{J}_\alpha\}]}{\delta \mathbf{J}_\alpha(\mathbf{r}, t)} = 0, \quad (\min) \tag{25}$$

where $R_t[\{\rho_\alpha, \mathbf{J}_\alpha\}]$ is minimal at the solution $\mathbf{J}_\alpha(\mathbf{r}, t)$ to this implicit equation [52, 53, 66]. The functional consists of physically distinct contributions according to

$$R_t[\{\rho_\alpha, \mathbf{J}_\alpha\}] = \dot{F}[\{\rho_\alpha\}] + \sum_\alpha P_t^{\mathrm{id}}[\rho_\alpha, \mathbf{J}_\alpha] + P_t^{\mathrm{exc}}[\{\rho_\alpha, \mathbf{J}_\alpha\}] - \sum_\alpha X_t[\rho_\alpha, \mathbf{J}_\alpha]. \tag{26}$$

These contributions comprise the time derivative of the free energy functional (22)

$$\dot{F}[\{\rho_\alpha\}] = \sum_\alpha \int d\mathbf{r} \, \mathbf{J}_\alpha(\mathbf{r}, t) \cdot \nabla \frac{\delta F[\{\rho_\alpha\}]}{\delta \rho_\alpha(\mathbf{r}, t)}, \tag{27}$$

the ideal dissipation functional

$$P_t^{\mathrm{id}}[\rho_\alpha, \mathbf{J}_\alpha] = \frac{\gamma_\alpha}{2} \int d\mathbf{r} \frac{\mathbf{J}_\alpha^2(\mathbf{r}, t)}{\rho_\alpha(\mathbf{r}, t)}, \tag{28}$$

and the external power $X_t$ due to the external potential $V_{\mathrm{ext},\alpha}(\mathbf{r}, t)$ and external forces $\mathbf{f}_{\mathrm{ext},\alpha}(\mathbf{r}, t) = -\nabla V_{\mathrm{ext},\alpha}(\mathbf{r}, t) + \mathbf{f}_{\mathrm{nc},\alpha}(\mathbf{r}, t)$, with non-conservative external forces $\mathbf{f}_{\mathrm{nc},\alpha}(\mathbf{r}, t)$, given by

$$X_t[\rho_\alpha, \mathbf{J}_\alpha] = \int d\mathbf{r} \left( \mathbf{J}_\alpha(\mathbf{r}, t) \cdot \mathbf{f}_{\mathrm{ext},\alpha}(\mathbf{r}, t) - \rho_\alpha(\mathbf{r}, t) \dot{V}_{\mathrm{ext},\alpha}(\mathbf{r}, t) \right). \tag{29}$$

The genuine nonequilibrium contributions in (26) are contained in the superadiabatic excess free power functional $P_t^{\mathrm{exc}}[\{\rho_\alpha, \mathbf{J}_\alpha\}]$. Thus, from inserting (26) into (25), we obtain the Euler-Lagrange equation [53]

$$\gamma \frac{\mathbf{J}_\alpha(\mathbf{r}, t)}{\rho_\alpha(\mathbf{r}, t)} = \gamma \mathbf{v}_\alpha(\mathbf{r}, t) = -\nabla \frac{\delta F[\{\rho_\alpha\}]}{\delta \rho_\alpha(\mathbf{r}, t)} - \frac{\delta P_t^{\mathrm{exc}}[\{\rho_\alpha, \mathbf{J}_\alpha\}]}{\delta \mathbf{J}_\alpha(\mathbf{r}, t)} + \mathbf{f}_{\mathrm{ext}}(\mathbf{r}, t). \tag{30}$$

The superadiabatic free power functional generates the superadiabatic interparticle force field via

$$\mathbf{f}_{\text{sup},\alpha}(\mathbf{r},t) = -\frac{\delta P_t^{\text{exc}}[\{\rho_\alpha,\mathbf{J}_\alpha\}]}{\delta \mathbf{J}_\alpha(\mathbf{r},t)}. \tag{31}$$

Setting $P_t^{\text{exc}} = 0$, the PFT equation of motion (30) reduces to the DDFT (21).

Just like $F_{\text{exc}}$ in DFT, $P_t^{\text{exc}}$ is not known exactly and must be approximated in practice. We recently presented an approximation for superadiabatic forces in the dynamics of the van Hove function [57]. Here, the superadiabatic free power functional consists of three contributing functionals

$$P_t^{\text{exc}}[\rho_s,\rho_d,\mathbf{J}_s,\mathbf{J}_d] = P_t^{\text{visc}}[\rho,\mathbf{J}] + P_t^{\text{struc}}[\rho,\mathbf{J}] + P_t^{\text{drag}}[\rho_s,\rho_d,\mathbf{v}_\Delta], \tag{32}$$

where $P_t^{\text{visc}}[\rho,\mathbf{J}]$, $P_t^{\text{struc}}[\rho,\mathbf{J}]$ and $P_t^{\text{drag}}[\rho_s,\rho_d,\mathbf{v}_\Delta]$ are the contributions due to viscoelasticity, structural forces, and drag, respectively. The latter functional depends on the species-labelled density profiles $\rho_\alpha(\mathbf{r},t)$ and on the difference of the species-labelled velocity fields $\mathbf{v}_\Delta(\mathbf{r},t) = \mathbf{v}_s(\mathbf{r},t) - \mathbf{v}_d(\mathbf{r},t)$. The former two functionals depend only on the total current $\mathbf{J}(\mathbf{r},t) = \mathbf{J}_s(\mathbf{r},t) + \mathbf{J}_d(\mathbf{r},t)$ and on the total density profile $\rho(\mathbf{r},t) = \rho_s(\mathbf{r},t) + \rho_d(\mathbf{r},t)$. Each of these contributions is based on functionals developed previously for repulsive spheres in nonequilibrium situations [58, 59, 75, 76]. We describe the mathematical structure of each functional in the following.

The viscoelastic contribution is based on the velocity gradient functional presented by de las Heras and Schmidt [75]. Treffenstädt and Schmidt extended this functional with a concrete approximate form for the memory kernel [58] in a study of the hard sphere liquid under inhomogeneous shear. They showed that this functional describes very accurately viscoelastic effects in the sheared hard sphere system. The general viscoelastic functional form is given by

$$P_t^{\text{visc}}[\rho,\mathbf{J}] = \int d\mathbf{r} \int d\mathbf{r}' \int_{-\infty}^{t} dt' \rho(\mathbf{r},t)\big[\eta(\nabla \times \mathbf{v})\cdot(\nabla' \times \mathbf{v}') + \zeta(\nabla \cdot \mathbf{v})(\nabla' \cdot \mathbf{v}')\big]$$
$$\times \rho(\mathbf{r}',t')K(\mathbf{r}-\mathbf{r}',t-t'), \tag{33}$$

where $\mathbf{v} \equiv \mathbf{v}(\mathbf{r},t)$, $\mathbf{v}' \equiv \mathbf{v}(\mathbf{r}',t')$ is a shorthand notation and $\eta$ and $\zeta$ are constants. Here, $K(\mathbf{r},t)$ is a memory kernel, which describes the interaction of the system with its own past, given by

$$K(\mathbf{r},t) = \frac{e^{-\mathbf{r}^2/(4D_M t)-t/\tau_M}}{(4\pi D_M t)^{3/2}\tau_M}\Theta(t), \tag{34}$$

with memory time $\tau_M$ and memory diffusion constant $D_M$. (For simplicity, we assume that $K(\mathbf{r},t)$ is identical for the rotational and divergence contributions in (33).) In the case of the bulk van Hove function, $P_t^{\text{visc}}[\rho,\mathbf{J}]$ simplifies, since $\nabla \times \mathbf{v} = 0$ due to symmetry. In addition, we replace the local density $\rho(\mathbf{r},t)$ by the weighted density $n_3(\mathbf{r},t)$ of FMT, to better capture packing effects. The weighted density $n_3(\mathbf{r},t)$ is calculated by a convolution of the local density $\rho(\mathbf{r},t)$ with a weight function

$$\omega_3(\mathbf{r}) = \Theta(\sigma/2 - |\mathbf{r}|), \tag{35}$$

where $\Theta(\cdot)$ is the Heaviside step function. From (33), we hence obtain the explicit functional form

$$P_t^{\text{visc}}[\rho,\mathbf{J}] = \frac{C_{\text{visc}}}{2}\int d\mathbf{r}\int d\mathbf{r}'\int_0^t dt' n_3(\mathbf{r},t)(\nabla \cdot \mathbf{v})(\nabla' \cdot \mathbf{v}')n_3(\mathbf{r}',t')K(\mathbf{r}-\mathbf{r}',t-t'), \tag{36}$$

where $C_{\text{visc}}$ is a constant that determines the overall strength.

The drag contribution in (32) was originally proposed by Krinninger *et al.* [83, 84] and subsequently used by Geigenfeind *et al.* [59], who studied a binary mixture of monodisperse hard spheres that are driven against each other and hence display nonequilibrium lane formation [85]. They identified a drag force between the two species of the liquid. Here, we use their functional to approximate the drag force between the test particle and the distinct particles

$$P_t^{\text{drag}}[\rho_s, \rho_d, \mathbf{v}_\Delta] = \frac{C_{\text{drag}}}{2} \int d\mathbf{r} \, \rho_s(\mathbf{r}, t) \rho_d(\mathbf{r}, t) \mathbf{v}_\Delta^2(\mathbf{r}, t), \tag{37}$$

with prefactor $C_{\text{drag}}$. The drag functional (37) models the friction between the different particle species, when they move in opposite directions.

Geigenfeind *et al.* [59] also proposed two *structural* functionals. The respective resulting structural forces create inhomogeneity in the density profile of a nonequilibrium system [59, 76, 79]. We apply a structural force term similar to Eq. (51) in [59]:

$$P_t^{\text{struc}}[\rho, \mathbf{J}] = -C_{\text{struc}} \int d\mathbf{r} \int d\mathbf{r}' \int_0^t dt' [\nabla \cdot \mathbf{J}(\mathbf{r}, t)] K(\mathbf{r} - \mathbf{r}', t - t') n_3^2(\mathbf{r}', t') \mathbf{v}^2(\mathbf{r}', t'), \tag{38}$$

where $C_{\text{struc}}$ is a prefactor, and $K(\mathbf{r}, t)$ is a memory kernel of the form (34), but with different parameters $D_M$ and $\tau_M$ as used in the viscoelastic functional (36). In contrast to ref. [59] here we use the total instead of the differential velocity profile.

The functionals (36) - (38) and the corresponding superadiabatic forces that result from functional differentiation according to (31) fall into two different categories, as characterised by the symmetry between the force fields acting on the self and distinct density components, respectively. In the category of *total forces*, the force fields $\mathbf{f}_s(\mathbf{r}, t)$ and $\mathbf{f}_d(\mathbf{r}, t)$ acting on the self and distinct density components, respectively, are identical: $\mathbf{f}_s(\mathbf{r}, t) = \mathbf{f}_d(\mathbf{r}, t)$. This means that the force field from these functionals does not depend on species-labelled information, but rather depends only on the total density and current in the test particle system.

On the other hand, in the category of *differential forces*, the force densities $\mathbf{F}_s(\mathbf{r}, t)$ and $\mathbf{F}_d(\mathbf{r}, t)$ acting on the two density components are equal in magnitude, but opposite in direction: $\mathbf{F}_s(\mathbf{r}, t) = -\mathbf{F}_d(\mathbf{r}, t)$. The two categories were developed by Geigenfeind *et al.* for a binary colloidal system [59], but they are general and hence apply in the context of the van Hove function as well.

For $P_t^{\text{drag}}[\rho, \mathbf{J}]$, the force density components satisfy the relation

$$\begin{aligned} \mathbf{F}_s^{\text{drag}}(\mathbf{r}, t) &\equiv -\frac{\delta P_t^{\text{drag}}[\rho_s, \rho_d, \mathbf{v}_\Delta]}{\delta \mathbf{v}_s(\mathbf{r}, t)} \\ &= -C_{\text{drag}} \rho_s(\mathbf{r}, t) \rho_d(\mathbf{r}, t) \mathbf{v}_\Delta(\mathbf{r}, t) \\ &= -\mathbf{F}_d^{\text{drag}}(\mathbf{r}, t). \end{aligned} \tag{39}$$

Thus, the drag functional falls into the *differential* category, and the total drag force density vanishes,

$$\mathbf{F}_{\text{sup}}^{\text{drag}}(\mathbf{r}, t) = \mathbf{F}_s^{\text{drag}}(\mathbf{r}, t) + \mathbf{F}_d^{\text{drag}}(\mathbf{r}, t) = 0. \tag{40}$$

This implies that there is no total drag force due to this functional. (See Refs. [82, 86] for exact force sum rules that stem from Noether's Theorem.)

Geigenfeind *et al.* [59] defined the differential force density $\mathbf{G}(\mathbf{r}, t)$ as a linear combination of species-resolved force densities

$$\mathbf{G}(\mathbf{r}, t) \equiv \frac{\rho_d(\mathbf{r}, t)}{\rho(\mathbf{r}, t)} \mathbf{F}_s(\mathbf{r}, t) - \frac{\rho_s(\mathbf{r}, t)}{\rho(\mathbf{r}, t)} \mathbf{F}_d(\mathbf{r}, t). \tag{41}$$

(Note that $\mathbf{G}(\mathbf{r}, t)$ is not to be confused with the van Hove function $G_\alpha(\mathbf{r}, t)$, $\alpha = \mathrm{s}, \mathrm{d}$.) In the case of the drag functional (37) we obtain (41) explicitly as

$$\mathbf{G}^{\mathrm{drag}}(\mathbf{r}, t) = C_{\mathrm{drag}} \frac{\rho_{\mathrm{s}}(\mathbf{r}, t)\rho_{\mathrm{d}}(\mathbf{r}, t)}{\rho(\mathbf{r}, t)} \mathbf{v}_\Delta(\mathbf{r}, t)\rho_\Delta(\mathbf{r}, t), \tag{42}$$

with differential density $\rho_\Delta(\mathbf{r}, t) = \rho_{\mathrm{s}}(\mathbf{r}, t) - \rho_{\mathrm{d}}(\mathbf{r}, t)$.

In contrast, the force fields generated by the functionals $P_t^{\mathrm{visc}}[\rho, \mathbf{J}]$ and $P_t^{\mathrm{struc}}[\rho, \mathbf{J}]$ fall into the category of total forces. Therefore, the viscoelastic force density $\mathbf{F}_\alpha^{\mathrm{visc}}(\mathbf{r}, t)$ for the component $\alpha = \mathrm{s}, \mathrm{d}$ is

$$\mathbf{F}_\alpha^{\mathrm{visc}}(\mathbf{r}, t) = -\rho_\alpha(\mathbf{r}, t) \frac{\delta P_t^{\mathrm{visc}}[\rho, \mathbf{J}]}{\delta \mathbf{J}(\mathbf{r}, t)} \equiv \rho_\alpha(\mathbf{r}, t)\mathbf{f}^{\mathrm{visc}}(\mathbf{r}, t), \tag{43}$$

where $\mathbf{f}^{\mathrm{visc}}(\mathbf{r}, t)$ is species-independent. The differential force density due to the viscoelastic functional (36) vanishes,

$$\begin{aligned}
\mathbf{G}^{\mathrm{visc}}(\mathbf{r}, t) &= \frac{\rho_{\mathrm{d}}(\mathbf{r}, t)}{\rho(\mathbf{r}, t)} \mathbf{F}_{\mathrm{s}}^{\mathrm{visc}}(\mathbf{r}, t) - \frac{\rho_{\mathrm{s}}(\mathbf{r}, t)}{\rho(\mathbf{r}, t)} \mathbf{F}_{\mathrm{d}}^{\mathrm{visc}}(\mathbf{r}, t) \\
&= \frac{\rho_{\mathrm{s}}(\mathbf{r}, t)\rho_{\mathrm{d}}(\mathbf{r}, t)}{\rho(\mathbf{r}, t)} \left( \mathbf{f}^{\mathrm{visc}}(\mathbf{r}, t) - \mathbf{f}^{\mathrm{visc}}(\mathbf{r}, t) \right) = 0,
\end{aligned} \tag{44}$$

and the same holds true for the structural superadiabatic force density that is generated from the functional (38). Thus, splitting of superadiabatic forces into total force $\mathbf{f}_{\mathrm{sup}}(\mathbf{r}, t)$ and differential force density $\mathbf{G}_{\mathrm{sup}}(\mathbf{r}, t)$, instead of using the species-resolved force density $\mathbf{F}_{\mathrm{s}}(\mathbf{r}, t)$ and $\mathbf{F}_{\mathrm{d}}(\mathbf{r}, t)$, is helpful to identify the underlying physics of the different force terms. Summarising, we obtain within our approximation the total force

$$\mathbf{f}_{\mathrm{sup}}(\mathbf{r}, t) = -\frac{\delta P_t^{\mathrm{visc}}[\rho, \mathbf{J}]}{\delta \mathbf{J}(\mathbf{r}, t)} - \frac{\delta P_t^{\mathrm{struc}}[\rho, \mathbf{J}]}{\delta \mathbf{J}(\mathbf{r}, t)} \tag{45}$$

$$= \frac{C_{\mathrm{visc}}}{2\rho(\mathbf{r}, t)} \int \mathrm{d}\mathbf{r}' \int_0^t \mathrm{d}t' n_3' \left( \nabla' \cdot \mathbf{v}' \right) \nabla (n_3 K) - C_{\mathrm{struc}} \int \mathrm{d}\mathbf{r}' \int_0^t \mathrm{d}t' n_3' \mathbf{v}'^2 \nabla K, \tag{46}$$

where $n_3' \equiv n_3(\mathbf{r}', t')$, $n_3 \equiv n_3(\mathbf{r}, t)$ and $K \equiv K(\mathbf{r} - \mathbf{r}', t - t')$ is a shorthand notation, and the differential force density

$$\mathbf{G}_{\mathrm{sup}}(\mathbf{r}, t) = -\frac{\delta P_t^{\mathrm{drag}}[\rho_{\mathrm{s}}, \rho_{\mathrm{d}}, \mathbf{v}_\Delta]}{\delta \mathbf{v}_\Delta(\mathbf{r}, t)} = \mathbf{G}^{\mathrm{drag}}(\mathbf{r}, t), \tag{47}$$

with $\mathbf{G}^{\mathrm{drag}}(\mathbf{r}, t)$ as defined in (42).

## 2.6 Long-Time Self Diffusion

At long times $t \gg \tau$, much of the structure of the van Hove function disappears. To obtain a simplified approximation for the long-time decay of the van Hove function, we make the following assumptions, which shall be supported below by observations made in BD simulation (see Sec. 3.2 and 3.3). We take the self density profile to be a Gaussian that diffuses in time with the long-time diffusion constant $D_{\mathrm{L}}$,

$$\rho_{\mathrm{s}}(\mathbf{r}, t) = (4\pi D_{\mathrm{L}} t)^{-3/2} \exp\left( -\frac{r^2}{4 D_{\mathrm{L}} t} \right), \tag{48}$$

where the value of $D_L$ is yet to be determined. The density profile that represents the total van Hove function is assumed to be uniform and constant,

$$\rho(\mathbf{r}, t) \equiv \rho_s(\mathbf{r}, t) + \rho_d(\mathbf{r}, t) = \rho_B, \tag{49}$$

hence any deviations from this infinite time behaviour are neglected. Under the assumption (49), the adiabatic force field vanishes everywhere. By inverting the spatial derivative in the continuity equation (14), we can identify the self velocity field that corresponds to (48) as

$$\mathbf{v}_s(\mathbf{r}, t) = \frac{r}{2t}\hat{\mathbf{e}}_r, \tag{50}$$

where $\hat{\mathbf{e}}_r$ is the radial unit vector. From (49) it follows further that the self and distinct current fields are equal in magnitude and opposite in direction, $\mathbf{J}_s(\mathbf{r}, t) = -\mathbf{J}_d(\mathbf{r}, t)$ and thus the total current vanishes,

$$\mathbf{J}(\mathbf{r}, t) = 0. \tag{51}$$

Hence, the viscoelastic (36) and structural superadiabatic forces (38), which depend only on the total density and total current, vanish as well. The only remaining forces are the ideal force, which drives the long-time diffusion, and the superadiabatic drag force, which slows this process down.

The PFT Euler-Lagrange equation (30) hence simplifies under the above assumptions to

$$\frac{\delta P_t^{id}[\mathbf{J}_s]}{\delta \mathbf{J}_s(\mathbf{r}, t)} + \nabla \frac{\delta F_{id}[\rho_s]}{\delta \rho_s(\mathbf{r}, t)} + \frac{\delta P_t^{drag}[\rho_s, \rho_d, \mathbf{v}_\Delta]}{\delta \mathbf{J}_s(\mathbf{r}, t)} = 0. \tag{52}$$

We show in the following that (52) can be solved analytically. After plugging in the above forms of the density field (48), the self velocity (50) and the current field (51), we obtain for the ideal dissipation functional given by (28)

$$\frac{\delta P_t^{id}[\mathbf{J}_s]}{\delta \mathbf{J}_s(\mathbf{r}, t)} \equiv \gamma \mathbf{v}_s(\mathbf{r}, t) = \gamma \frac{r}{2t}\hat{\mathbf{e}}_r, \tag{53}$$

then for the ideal diffusion functional

$$\nabla \frac{\delta F_{id}[\rho_s]}{\delta \rho_s(\mathbf{r}, t)} \equiv k_B T \nabla \ln\left(\Lambda^3 \rho_s(\mathbf{r}, t)\right) = -k_B T \frac{r}{2D_L t}\hat{\mathbf{e}}_r, \tag{54}$$

and finally for the superadiabatic drag force

$$\begin{aligned}
\frac{\delta P_t^{drag}[\rho_s, \rho_d, \mathbf{v}_\Delta]}{\delta \mathbf{J}_s(\mathbf{r}, t)} &= C_{drag}\rho_d(\mathbf{r}, t)(\mathbf{v}_s(\mathbf{r}, t) - \mathbf{v}_d(\mathbf{r}, t)) \\
&= C_{drag}(\mathbf{v}_s(\mathbf{r}, t)\rho_d(\mathbf{r}, t) - \mathbf{J}_d(\mathbf{r}, t)) \\
&= C_{drag}(\mathbf{v}_s(\mathbf{r}, t)\rho_d(\mathbf{r}, t) + \mathbf{J}_s(\mathbf{r}, t)) \\
&= C_{drag}(\mathbf{v}_s(\mathbf{r}, t)\rho_d(\mathbf{r}, t) + \mathbf{v}_s(\mathbf{r}, t)\rho_s(\mathbf{r}, t)) \\
&= C_{drag}\mathbf{v}_s(\mathbf{r}, t)(\rho_d(\mathbf{r}, t) + \rho_s(\mathbf{r}, t)) \\
&= C_{drag}\rho_B \mathbf{v}_s(\mathbf{r}, t) \\
&= C_{drag}\rho_B \frac{r}{2t}\hat{\mathbf{e}}_r.
\end{aligned} \tag{55}$$

A comparison of coefficients in (52) leads to the relation

$$\gamma - \frac{k_B T}{D_L} + C_{drag}\rho_B = 0, \tag{56}$$

which results in a modified Einstein relation for the long-time self diffusion in an interacting system given by

$$D_{\mathrm{L}} = \frac{\mathrm{k_B}T}{\gamma + C_{\mathrm{drag}}\rho_{\mathrm{B}}} \, . \tag{57}$$

Below we test the approximation (57) by comparing the resulting behaviour of the diffusion coefficient $D_{\mathrm{L}}$ with simulation results.

The decrease of $D_{\mathrm{L}}$ that results from a finite value $C_{\mathrm{drag}} > 0$ according to equation (57) is an entirely superadiabatic effect. Going through the above derivation on the basis of the DDFT alone leads to the trivial result $D_{\mathrm{L}} = \mathrm{k_B}T/\gamma$, analogous to formally setting $C_{\mathrm{drag}} = 0$ in equation (57). Hence the power functional ansatz for $P_t^{\mathrm{drag}}$, see equations (37) and (55), links drag (due to interflow of the self and distinct components) with the long time self diffusion coefficient.

## 2.7 Simulation Parameters

We simulate $N = 1090$ hard spheres in a box of size $10 \times 10 \times 15 \ \sigma^3$ with periodic boundary conditions, resulting in a bulk density $\rho_{\mathrm{B}} \approx 0.73\sigma^{-3}$ and a packing fraction $\eta \approx 0.38$. The same parameters were used in [57]. We take $10^6$ simulation snapshots over a simulation time of $10^3\tau$ to calculate observables.

Simulations of the Lennard-Jones liquid were carried out with $N = 500$ particles at mean number density $\rho = 0.84\sigma^{-3}$ and absolute temperature $\mathrm{k_B}T = 0.8\epsilon$ in a box with periodic boundary conditions, where $\epsilon$ sets the energy scale of the Lennard-Jones interparticle interaction potential

$$\Phi_{\mathrm{LJ}}(r) = 4\epsilon\left[\left(\frac{\sigma}{r}\right)^{12} - \left(\frac{\sigma}{r}\right)^6\right], \tag{58}$$

with particle distance $r$. The potential is truncated at $r_{\mathrm{c}} = 4\sigma$.

The DDFT calculations are carried out in a spherical box with radius $32\sigma$ and a radial discretisation step of $\Delta r = 2^{-7}\sigma = 7.8125 \cdot 10^{-3}\sigma$, beyond which both density components are continued as a constant. The integration time step is $10^{-6}\tau$. The code is available online [87].

# 3 Results

## 3.1 Radial Distribution Function

As a consistency check, we first calculate the equilibrium radial distribution function $g(r)$ from BD simulation data and compare the results with those obtained using DFT. The latter are used as an initial condition for our DDFT calculations. As described above, at $t = 0$ the distinct part of the van Hove function is equal to the radial distribution function $g(r) = G_d(r,0)/\rho_{\mathrm{B}}$. For hard spheres, $g(r)$ is zero for $r < \sigma$, which corresponds to the excluded volume around the test particle, see figure 1. Starting at $r = \sigma$ and going away from the origin, each subsequent local maximum of $g(r)$ corresponds to a correlation shell of particles around the test particle. The height of these maxima decays rapidly, with an exponential envelope, as the distance to the test particle increases. The asymptotic decay at large distances $r$ follows an exponentially damped oscillating law

$$\frac{r}{\sigma}(g(r) - 1) = \tilde{A}\mathrm{e}^{-\tilde{\alpha}_0 r}\cos(\tilde{\alpha}_1 r - \theta) \tag{59}$$

with amplitude $\tilde{A}$, inverse decay length $\tilde{\alpha}_0$, wave number $\tilde{\alpha}_1$ and phase shift $\theta$ [88]. When plotting $\ln(r\,|g(r) - 1|/\sigma)$ against $r$ (see figure 1 (b)), the exponential envelope is clearly

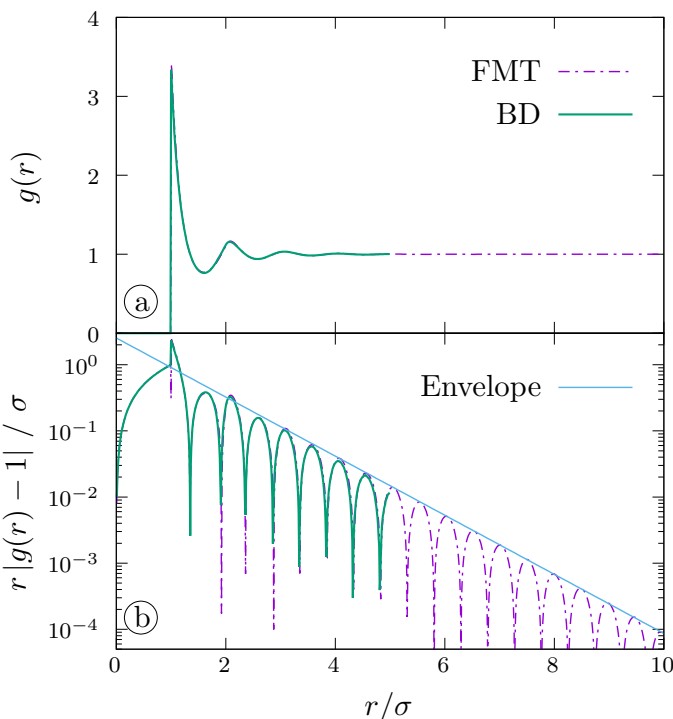

Figure 1: The radial distribution function $g(r)$ of a hard sphere liquid as a function of the distance $r/\sigma$ at packing fraction $\eta \approx 0.38$. Results from FMT (dash-dotted purple line) are compared to BD simulation (full green line). Panel (a) shows the radial distribution function on a linear scale. Panel (b) shows the deviation $\ln(r\,|g(r)-1|/\sigma)$ from the bulk value on a logarithmic scale. Here, many correlation shells can be seen. The envelope (thin blue line) of the maxima of the shells follows an exponential law.

identifiable in the diagram as a straight line along which a sequence of local maxima visibly aligns. We obtain values of $\tilde{\alpha}_0 \approx 1.0\sigma^{-1}$ and $\tilde{\alpha}_1 \approx 6.4\sigma^{-1}$. The correlation shells are spaced apart by a distance $2\pi/\tilde{\alpha}_1 \approx 1.0\sigma$. These values coincide closely with literature values $\tilde{\alpha}_0 \approx 0.97\sigma^{-1}$ and $\tilde{\alpha}_1 \approx 6.45\sigma^{-1}$ [88], obtained at a density of $\rho = 0.75\sigma^{-3}$, similar to our bulk density $\rho = 0.73\sigma^{-3}$.

The agreement between $g(r)$ obtained with BD simulation and obtained with FMT in the present test particle setup is excellent, as expected for the White Bear Mk. 2 excess free energy functional [38].

## 3.2 Dynamic Structural Decay

### 3.2.1 Self Correlations

The self part $G_s(r,t)$ of the van Hove function at $t = 0$ is equal to a Dirac delta distribution localised at the origin of the coordinate system, cf. Eq. (8). Over time, the distribution widens and it attains the shape of a bell curve (see figure 2). In a freely diffusing system without interparticle interactions, the density distribution is given by a Gaussian,

$$f_G(r,t) = (2\pi\sigma^2(t))^{-\frac{3}{2}} \exp\left(-\frac{r^2}{2\sigma^2(t)}\right), \tag{60}$$

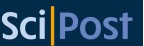

Figure 2: Self (yellow dashed lines) and distinct (solid blue lines) part of the van Hove correlation function $G(r, t)$ at different times $t$, with radial distribution function $g(r)$ (dotted grey lines) and Gaussian $f_G(r, \sigma(t))$ (dash-dotted red lines) with mean square displacement $\langle r^2 \rangle = 3\sigma^2$ fitted to match that of the self correlation. Results from BD (left column, panels (a) – (e)) are compared to DDFT with full and quenched excess free energy functional (middle column, panels (f) – (j)) and to DDFT with partially linearised excess free energy functional (right column, panels (k) – (o)). The maxima of the solid line correspond to the local extrema of the distinct van Hove function. The first maximum of $r |G_d(r, t)/\rho_B - 1|$ always corresponds to a minimum of $G_d(r, t)$.

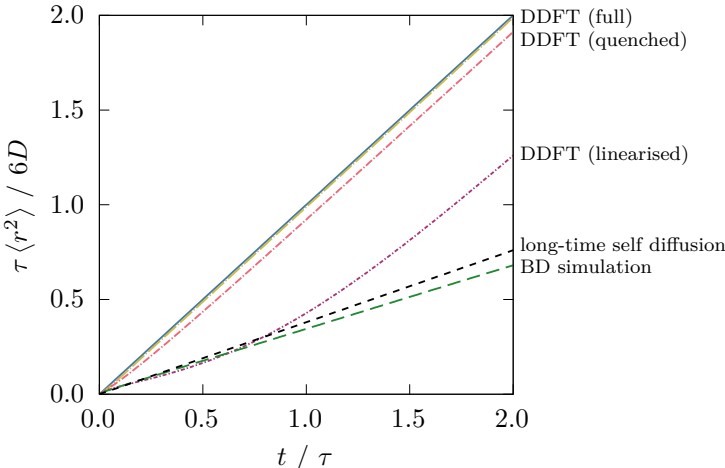

Figure 3: Mean square displacement $\langle r^2 \rangle \tau/6D$ as a function of time $t/\tau$ of the self component of the van Hove function. Data is shown from BD simulation (green long dashes) and from DDFT with the full (yellow long dot-dashes), quenched (red medium dot-dashes) and partially linearised (purple short dot-dashes) excess free energy functional. Additionally, we show the long-time self diffusion (black short dashes) as calculated from our PFT approximation (57). As a reference, the linear law of ideal diffusion (blue solid line) is also plotted.

with time-dependent variance $\sigma^2(t)$, for which the relation

$$\sigma^2(t) = 2Dt, \tag{61}$$

holds [3]. To relate the shape of $G_s(\mathbf{r}, t)$ in the interacting system to (60), we calculate the mean square displacement at time $t$ via

$$\langle r^2(t) \rangle = \int d\mathbf{r}\, r^2 G_s(r, t), \tag{62}$$

and compare to an equivalent Gaussian of the form (60) with variance chosen such that

$$\sigma^2(t) = \frac{1}{3} \langle r^2(t) \rangle. \tag{63}$$

We perform this analysis with data both from BD simulation and from DDFT calculations. For BD, the difference between $G_s(r, t)$ and the equivalent Gaussian is minimal, see figure 2 (a) – (e). In DDFT using either the full or the quenched functionals, although the shape of $G_s(r, t)$ is also Gaussian, see figure 2 (f) – (j), the resulting variance is significantly different from the BD results at the same time $t$. The full and the quenched functionals give very similar results, with no major improvement occurring upon using the quenched approach. When plotting the mean square displacement against time, we see that $G_s(r, t)$ as approximated by these functionals behaves much closer to ideal diffusion than to the BD simulation, see figure 3. We recall that this behaviour is consistent with the absence of superadiabatic effects in the DDFT.

The shape of the self van Hove function in DDFT using the partially linearised functional shows major deviations from a Gaussian, see figure 2 (k) – (o). The distribution is both more strongly localised at the origin and it has a significantly longer tail at large distances, compared to a Gaussian with identical mean square displacement. For $t \leq \tau$, the mean square displacement is much closer to BD simulation than with either the full or quenched functional, but it deviates strongly for $t > \tau$ with a slope approaching that of ideal diffusion, see figure 3.

While the mean square displacement as a metric might suggest otherwise, the deviations in the shape of $G_s(r,t)$ make the linearised modification unfit as an improvement over an approach that does not correct at all for self interactions.

### 3.2.2 Distinct Correlations

Figure 2 shows also a comparison of BD and DDFT results for the dynamic decay of the distinct part of the van Hove function. The temporal decay consists of two stages, namely of an initial deconfinement and of a subsequent outward drift of correlation shells: First, particles diffuse out of the high density regions into the neighbouring minima. Thus, both maxima and minima become less pronounced. However, this process is not equally rapid everywhere. Inner shells, i.e. those that are close to the test particle, decay at earlier times than do outer shells. As the test particle, whose confinement at the origin caused the appearance of the shell structure in the first place, diffuses away from the origin, the particles of the first correlation shell start to drift closer to the origin, which in turn allows the second shell to expand inwards, and so on for further shells. Thus, inner shells decay in an expanding region around the origin, while outer

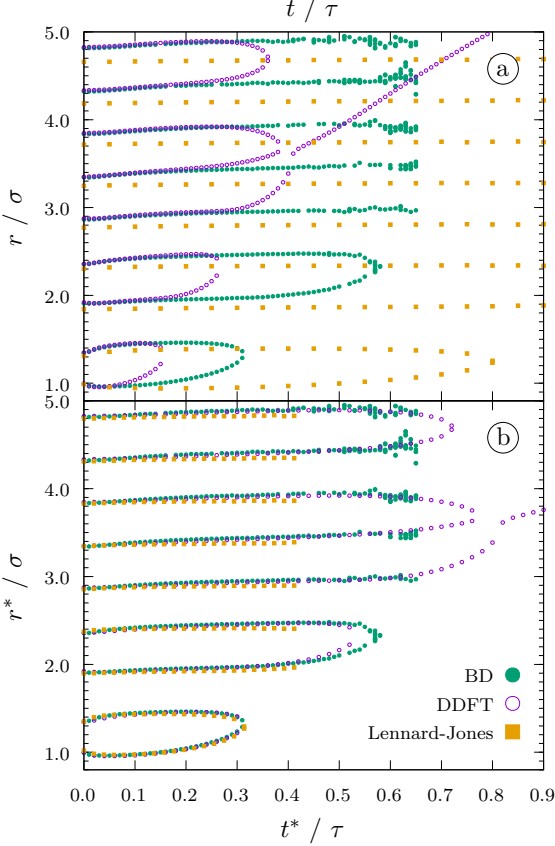

Figure 4: Zeros of $r\left(\frac{G_d(r,t)}{\rho_B} - 1\right)$ as a function of time. Data from BD simulation (green, suppressed for $t > 0.65\tau$ due to noise), DDFT with the quenched functional (purple), and results for the Lennard-Jones liquid (yellow). Panel (a) shows the data unmodified, whereas in panel (b), time $t$ and distance $r$ have been rescaled to allow for a comparison of the qualitative behaviour. Here the rescaled quantities are marked by an asterisk and the rescaling is such that $t = t^*$, $r = r^*$ for hard sphere BD simulation, $2t = t^*$ and $r = r^*$ for hard sphere DDFT, and $t/2.55 = t^*$ and $1.03r = r^*$ for the Lennard-Jones liquid.

shells remain stable for a longer time. This mechanism results in an additional lengthscale for the spatial decay of the inner shells, while the decay length of the outer shells remains equal to the static decay length of $g(r)$. This dynamic scenario has also been found by Schindler and Schmidt [51], who reported that for the Lennard-Jones liquid the new, dynamic decay length increased, starting from the static decay length, up until $t = 0.7\tau$, where it reached a plateau value. Here we find an increase of the dynamic decay length up until at least $t = 0.3\tau$, but cannot identify a plateau value, due to numerical noise and possibly a somewhat smaller system size.

The second stage of decay is caused by the test particle diffusing out of its initial position, thus creating space for distinct particles to come closer to the origin. The particles essentially flow into the initial cavity, which swallows correlation shells from the inside out. In this second regime, the distinct correlations decrease monotonically and faster than exponential with distance. This second stage of dynamic decay has not been explicitly mentioned in Ref. [51], but can be seen in figure 3 of that paper, where the first maximum of $G_d(r, t)$ decreases in amplitude and then melts into the minimum at the origin.

Additionally to the above effects, we observe an outward drift of the correlation shells, which seems to be hitherto unreported. We analyse the locations of the zeros of $r\left(\frac{G_d(r,t)}{\rho_B} - 1\right)$ as a function of time; results are shown in figure 4. We observe a slow, close to linear in time, drift of the inner zeros with a drift speed of $(0.24 \pm 0.01)\sigma/\tau$, up to the point where the corresponding shell is swallowed by the central cavity and the corresponding zeros disappear. By analysing $G_d(r, t)$ for the Lennard-Jones liquid at temperature $T = 0.8\epsilon/k_B$ and density $\rho = 0.84/\sigma_{LJ}^3$, we observe a similar, but much slower drift of $(0.037 \pm 0.001)\sigma/\tau$ (see figure 4). A rescaling of time and distance, shown in panel (b) of figure 4, shows that the behaviour of the zeros of the van Hove function of the Lennard-Jones liquid is qualitatively very similar to the behaviour of the hard sphere liquid. These findings suggest that the mechanism of dynamic decay presented here and by Schindler and Schmidt is, at least qualitatively, universal for particles with repulsive interactions and overdamped dynamics.

The behaviour described above is qualitatively reproduced by DDFT, based on either the full or the quenched approach. Both functionals yield results that well represent the shape of $G_d(\mathbf{r}, t)$, but overestimate the rate of temporal decay by roughly a factor of two (compare e.g. figure 2(d) and 2(h) and recall the time rescaling in figure 4(b)). The differences of results from the full and quenched approach are minor, though. In contrast, the shell deconfinement as predicted with the linearised functional is slower than in BD. This is despite the fact that the diffusion of the self correlation is faster than in BD. However, this may be due to the stable inner peak of the self correlation, which in turn could stabilise the shell structure. Additionally, the shape of $G_d(\mathbf{r}, t)$ shows intermittent behaviour which is not observed at all in the time evolution in BD, see e.g. figure 2(o). These results substantiate the finding that the partial linearisation approach is not well suited to fix the self interaction problem.

It should be noted that all of the three excess free energy functionals studied produce identical adiabatic forces on the distinct density component, given the same input density. Yet, the time evolution of the distinct density component is significantly different between the three cases, which must therefore be caused by the difference in the time evolution of the self density component. Thus, the correct handling of self-interactions of the self density is not a minor detail, but rather a central requirement for test particle DFT.

Lastly, we observe that at time $t \approx 0.4\tau$, all but one zero of $r\left(\frac{G_d(r,t)}{\rho_B} - 1\right)$ disappear in our DDFT results (see figure 4). This disappearance constitutes the crossover to the long time regime discussed in section 2.6. The negative deflection in $\rho_d$ at small $r$ is the hole occupied by the self particle. The positive value of $r(G_d(r, t)/\rho_B - 1)$ at distances larger than the zero is caused by adhesion of distinct particles around the self particle in the initial condition. It depends on the static structure factor of the fluid in sign and magnitude. Diffusion of this

excess part has been studied in a time-dependent setup [89, 90]. We neglect this effect in section 2.6 as it is smaller than the hole of the self particle by a factor of $\sim 40$.

### 3.2.3 Vineyard Approximation

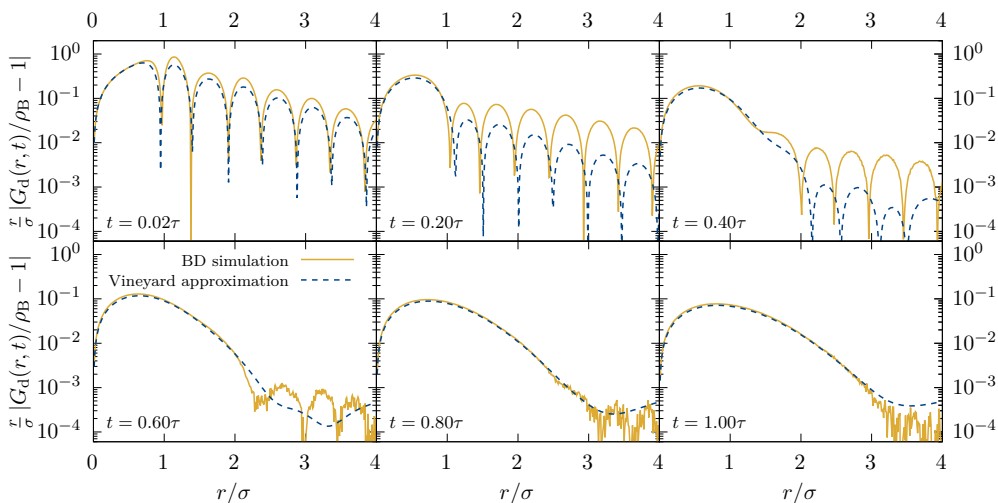

Figure 5: Comparison of the distinct van Hove function $G_{\mathrm{d}}(\mathbf{r}, t)$, obtained directly using BD simulation (solid yellow lines) and obtained from simulation data using the Vineyard approximation (dashed blue lines).

One early attempt at a theoretical approximation for the van Hove function was presented by Vineyard [3, 91]. He proposed the relation

$$G_{\mathrm{d}}(\mathbf{r}, t) \approx (g(\tilde{\mathbf{r}}) * G_{\mathrm{s}}(\tilde{\mathbf{r}}, t))(\mathbf{r}), \tag{64}$$

where the operator $*$ indicates a convolution, between the self component and the distinct component of the van Hove function. Hopkins *et al.* found the Vineyard approximation to be a 'fairly good approximation' of the van Hove function [47]. As a self-consistency check, we use our simulation data to calculate the convolution in equation (64) and compare the result to the simulation result for $G_{\mathrm{d}}(\mathbf{r}, t)$ (see figure 5). We find that the Vineyard approximation significantly overestimates the rate of decay of the shell structure of the distinct van Hove function, in accordance with previous findings [47]. Only the minimum at small distances $r$ is well reproduced. This indicates that, even if we had a perfect approximation of the behaviour of the self component of the van Hove function, we could not use the Vineyard approximation to obtain an equally accurate approximation of the distinct van Hove function.

## 3.3 Adiabatic Forces

We next analyse the forces that govern the time evolution of the van Hove function. The full internal force field for both the self and the distinct component can be calculated from BD simulation data from the species-resolved current profiles $\mathbf{J}_{\alpha}(\mathbf{r}, t)$ via (15). To split the internal force field into adiabatic and superadiabatic forces, we need to first calculate the adiabatic forces. In section 3.2, we have compared results from three different free energy functional approximations which produce adiabatic forces in the framework of DDFT. To test whether these functionals are accurate, we calculate the adiabatic potential

$$V_{\mathrm{ad}, \alpha}(\mathbf{r}, t) = \mu - \frac{\delta F[\rho_{\mathrm{s}}, \rho_{\mathrm{d}}]}{\delta \rho_{\alpha}(\mathbf{r}, t)}, \tag{65}$$

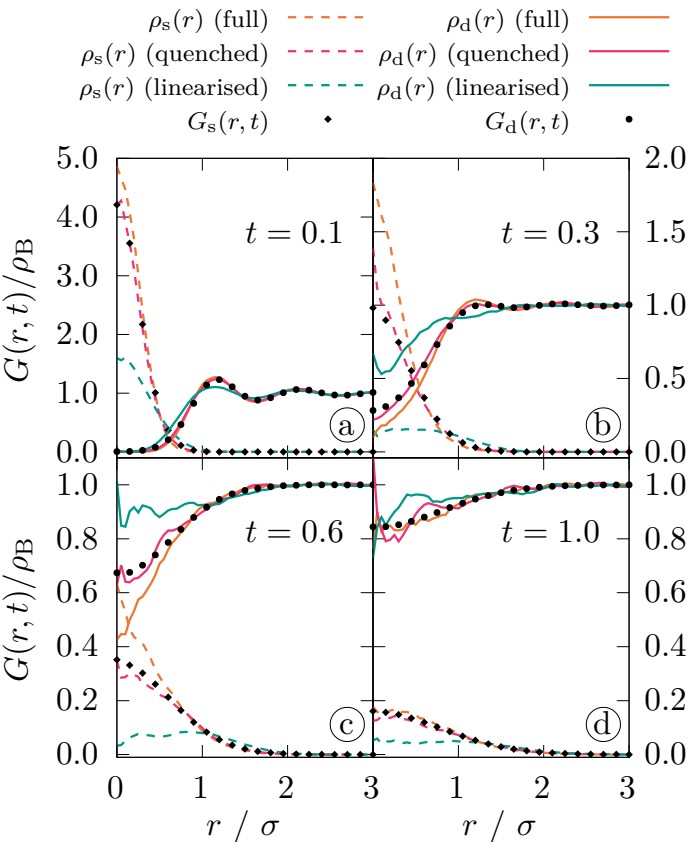

Figure 6: Self and distinct parts of the van Hove function $G_s(r, t)$ (black discs), $G_d(r, t)$ (black diamonds) as a function of distance $r$, from BD simulation, at times $t$ as indicated. Density profiles $\rho_\alpha(r)$ from test particle equilibrium BD simulation (self part: dashed lines, distinct part: solid lines) with adiabatic forces from DFT with full (orange), quenched (magenta) and partially linearised (teal) excess free energy.

using each of the three free energy functionals, with the density profiles sampled in BD simulation at times $t = 0.1\tau$, $0.3\tau$, $0.6\tau$ and $1.0\tau$. Then, we run equilibrium BD simulations with an external force field $\mathbf{f}_{\text{ext},\alpha}(\mathbf{r}, t) = -\nabla V_{\text{ad},\alpha}(\mathbf{r}, t)$ designed to cancel out the predicted adiabatic internal force field [65, 67]. We sample equilibrium density profiles from these simulations and compare them to the initial $G_\alpha(\mathbf{r}, t)$ profiles. If the given functional produces accurate adiabatic forces, then the resulting density profiles should be identical to the original input.

The results, shown in figure 6, indicate that both the full and the partially linearised excess free energy functional yield density profiles that significantly deviate from the target $G_\alpha(\mathbf{r}, t)$. For the quenched functional, the agreement is markedly improved, but relevant deviations remain near the origin. As we will see below, the forces calculated in this region, via the quenched functional, are still not reliable enough to facilitate an accurate splitting of the internal forces into adiabatic and superadiabatic contributions (see figure 7, panel b). In order to circumvent this problem, we iteratively calculate the species-resolved adiabatic external potential for the chosen times, using equilibrium Monte Carlo simulations. To ensure consistency, we choose the same number of particles and system dimensions as the BD simulations. The gradient of the calculated external potentials gives us the adiabatic force fields via (19), and by extension the superadiabatic forces via (17).

We examine the ideal, adiabatic and superadiabatic force fields as a function of time (see figure 7). The largest force contribution arises from the ideal diffusion, which always acts to smooth out any density inhomogeneity. Both the adiabatic and superadiabatic force mainly

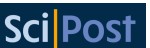

Figure 7: A comparison of forces and force densities from simulation (symbols) and DDFT/PFT approximation (lines). Lines and symbols coloured identically show the same measured quantity. **(a, b)**: Ideal (light blue, empty circles), adiabatic (green, squares) and superadiabatic (dark blue, solid circles) force density for the self (a) and distinct (b) component of the van Hove function as a function of distance $r$ for different times $t$, as measured in BD simulation. The quenched functional approximation for the adiabatic force density is shown as a solid green line. Shown as a dark blue solid line is the PFT approximation for the superadiabatic force density, consisting of a drag component (dashed yellow line) and the viscous plus structural component (dash-dotted red line). **(c)**: Splitting of superadiabatic forces into total force $\mathbf{f}_{\mathrm{sup}}$ (red diamonds) and differential force density $\mathbf{G}_{\mathrm{sup}}$ (yellow triangles) as a function of distance $r$ at different times $t$. We also show the different force components of our approximation, consisting of drag (dashed yellow line), viscosity (long-dashed purple line) and a structural force (dotted purple line). The drag component acts purely differentially, whereas viscosity and the structural component act purely as a total force.

oppose this relaxation process. Notably, at $t \geq \tau$ the adiabatic force field is small compared to both the ideal and the superadiabatic force field. Thus, DDFT, which by construction neglects superadiabatic contributions, must fail to describe the long-time behaviour of the van Hove function, even if adiabatic forces were included with perfect accuracy. This explains why the long-time diffusion of the self component tends to ideal diffusion for all studied DDFT approximations (see figure 3). Additionally, we can see in panel b of figure 7 that even the quenched approximation for the adiabatic force density shows some deviations in the distinct component near the origin. These deviations are of similar magnitude as the force itself and may thus not be neglected when trying to split the internal force field into adiabatic and superadiabatic contributions. However, we see that the distinct force density component for $r > \sigma$, as well as the self component, are accurately reproduced.

## 3.4 Superadiabatic Forces

In Sec. 2.5 we have described a kinematic approximation for the superadiabatic forces, that include three distinct physical effects: a viscoelastic force, a structural force and a drag force. With the density and current profiles from BD simulation as input, we can use this theory to calculate both the total force field and the differential force density field via (46) and (47). Crucially, the drag force (42) is the sole contribution to the differential superadiabatic force density $\mathbf{G}_{\mathrm{sup}}(\mathbf{r}, t)$, see (47). Recall that the drag force arises from the opposing motion of the two species.

In practice, we perform a least squares fit of the differential force density calculated using (47) against the corresponding measured BD simulation data. Thereby we identify the free parameter in Eq. (37) as $C_{\mathrm{drag}} \approx 2.2 k_{\mathrm{B}} T \tau \sigma$. The quantitative agreement with $\mathbf{G}_{\mathrm{sup}}(\mathbf{r}, t)$ is excellent for $t \geq 0.6\tau$, but minor deviations are apparent at earlier times. The fact that both the shape of $\mathbf{G}_{\mathrm{sup}}(\mathbf{r}, t)$ as well as the scaling with time are well represented by the drag functional constitutes strong evidence that drag is the relevant physical effect which governs the differential superadiabatic force density. The differential drag force density is quite simple in shape, making it almost a trivial contribution to the dynamics of the van Hove function, see figure 7. One could argue that the real complexity of the system therefore lies in the behaviour of the total van Hove function, and hence in the total force field.

We model the total force component based on the viscoelastic (36) and structural (38) functionals (see Eq. (46)). The parameters of the memory kernel in (36) were determined via an examination of a sheared system of hard spheres in [58], at identical bulk density, as $\tau_{\mathrm{M}}^{\mathrm{visc}} \approx 0.02\tau$ and $D_{\mathrm{M}}^{\mathrm{visc}} \approx 5.6\sigma^2/\tau$. We apply those same parameters here, see figure 7, and find that the viscoelastic functional produces a spatially oscillating force field, which represents the total force profile well for $r > 1.5\sigma$. This oscillation decays rather quickly in time compared to the other force components and it is lost in the statistical noise at $t > \tau$. We determine the prefactor of the viscoelastic functional via a least squares fit to BD simulation data and obtain $C_{\mathrm{visc}} \approx 5.8 k_{\mathrm{B}} T/(\tau\sigma^3)$. In addition to the oscillations of the viscoelastic force, the total force profile shows a larger peak at $r \approx \sigma$, which is produced in our approximation by the structural force. Since we have no prior information about this force, we determine its memory time $\tau_{\mathrm{M}}^{\mathrm{struc}} \approx 0.3\tau$ and the memory diffusion constant $D_{\mathrm{M}}^{\mathrm{struc}} \approx 0.35\sigma^2/\tau$, as well as the prefactor $C_{\mathrm{struc}} \approx 0.42 k_{\mathrm{B}} T \tau^2/\sigma$, via a least squares fit to BD results. The obtained memory time is larger than that for the viscoelastic force by more than an order of magnitude. Therefore, the structural force is much more long-lived than the viscoelastic force and persists, though with decreased amplitude, even at $t = \tau$.

Together, the sum of the structural and the viscoelastic force is in very good agreement with the total force profile from simulation for times $t > 0.3\tau$. For earlier times $t < 0.3\tau$, quantitative deviations occur, but the spatial shape is still qualitatively reproduced. The deviation between our power functional approximation and the forces in simulation for early

times are most likely a result of both the extreme inhomogeneity of the density profile at early times, as well as of the large local velocities that occur in this regime.

We noted above that the adiabatic force field for $t > \tau$ is small compared to both the superadiabatic and the ideal force field, see also figure 7. The superadiabatic force field is dominated by the drag force at $t > \tau$. This situation allows us to predict the long-time diffusion constant of the self peak of the van Hove function via the modified Einstein relation (57), since the assumptions made in Sec. 2.6 hold to a large degree. For our system, we obtain $D_{\mathrm{L}} = \frac{k_{\mathrm{B}}T}{\gamma + C_{\mathrm{drag}}\rho_{\mathrm{B}}} \approx 0.38\sigma^2/\tau$, using the value given above for $C_{\mathrm{drag}} = 2.2 k_{\mathrm{B}}T\tau\sigma$. We determine the long-time diffusion coefficient from the asymptotic slope of the mean square displacement of the self density profile in BD simulation and obtain $D_{\mathrm{L}} \approx 0.32\sigma^2/\tau$, which corresponds to a value of $C_{\mathrm{drag}} \approx 2.9 k_{\mathrm{B}}T\tau\sigma$. These results are in reasonable agreement (see also figure 3), but indicate that our value for $C_{\mathrm{drag}} \approx 2.2 k_{\mathrm{B}}T\tau\sigma$ obtained above might be an underestimate. Fitting the drag amplitude only to the largest times results in $C_{\mathrm{drag}} \approx 2.4 k_{\mathrm{B}}T\tau\sigma$, from which we obtain $D_{\mathrm{L}} \approx 0.36\sigma^2/\tau$, improving the match with the simulation result. This finding indicates that the differential force field contains contributions for early times $t < \tau$ which are not captured well by our approximation for the drag force.

Hence the results shown in figure 7 demonstrate explicitly the perhaps expected shortcoming of the DDFT that it does not capture the full dynamics of the system. Here the van Hove function plays a dual role in that it is both an equilibrium dynamical correlation function as well as a specific nonequilibrium temporal process. The latter is specified by the dynamical test particle limit and we recall our description of how this approach allows one to identify two- with one-body correlation functions in section 2.3. As the DDFT does not provide the full nonequilibrium dynamics on the one-body level, by extension it also fails to describe all forces that govern the time evolution of the van Hove function. These findings are consistent with the nonequilibrium Ornstein-Zernike framework [41, 42], where both adiabatic and superadiabatic direct correlation functions occur, and only the former are generated from the free energy density functional. The superadiabatic time direct correlation functions are genuine dynamical objects. The present study sheds light on this issue in the test particle picture.

## 4 Conclusion and outlook

We have studied the van Hove correlation function of the Brownian hard sphere liquid using BD simulations. We have also calculated the van Hove function using test particle DDFT and analysed the interparticle force field using PFT. Our analysis of the dynamic decay of the distinct van Hove function shows a two-stage process. The initial deconfinement of correlation shells leads to an intermediate dynamic decay length, which is followed by a monotonic spatially super-exponential decay as the self component mixes with the distinct component. Additionally, correlation shells drift slowly outward from the origin. A comparison with results for the overdamped Lennard-Jones liquid shows that, qualitatively, the same effects occur in both systems. This, together with the fact that the behaviour of the hard sphere liquid is prototypical for a wide range of liquids, indicates that these results reflect fundamental effects in the dynamics of the liquid state.

We have discussed the accuracy of the Vineyard approximation on the basis of our simulation data. Whether the Vineyard approximation can form a useful ingredient in the study of superadiabatic effects in the van Hove function remains to be seen.

We have analysed adiabatic forces in the dynamic decay of the van Hove correlation function using Monte Carlo simulation and the adiabatic construction. This analysis showed that the quenched excess free energy functional presented by Stopper *et al.* [56] is the best currently available approximation for adiabatic forces in the test particle picture. The remaining

deviations are however severe enough to warrant further development of the theory. One possible path for investigation is canonical decomposition as presented by de las Heras and Schmidt [92]. On the conceptual level, investigating the relationship of the theory of ref. [56] to the DFT for quenched-annealed mixtures [93–96] would be interesting.

We have isolated superadiabatic forces, which showed that, even assuming very accurate approximation of adiabatic forces, DDFT is inadequate to quantitatively describe the dynamic decay of the van Hove function, since superadiabatic forces play a major role in the time evolution of both the self and the distinct density components. The long-time behaviour of the DDFT approximation for $G_s(\mathbf{r}, t)$ approaches ideal diffusion, since adiabatic forces vanish at long times.

Using PFT, we have demonstrated the splitting of the superadiabatic force into a viscoelastic force, a drag force between the two different components of the van Hove function, and a structural force. These three force contributions dominate the superadiabatic force field for $t > 0.3\tau$. Approximations for these forces were previously developed for nonequilibrium dynamics. Their occurrence in the dynamics of the van Hove function shows a deep connection between nonequilibrium forces and equilibrium dynamics. The differential force density acting between the two components of the van Hove function is much simpler and easier to approximate than are the forces governing the evolution of the total van Hove function. In the long-time tail of the van Hove function, adiabatic contributions to the interparticle force field vanish, as do the viscoelastic and structural superadiabatic force contributions. Thus, the drag force determines the slowing-down of the long-time self diffusion of the hard sphere liquid. The long-time self diffusion constant calculated using our approximation is consistent with our direct BD simulation results.

Overall, our approximation has seven free parameters. Out of these, two, namely the memory time and memory diffusion constant of the viscoelastic force, have been determined previously for a system of hard spheres under a shear force [58]. The remaining five parameters have been determined via a least-squares fit to simulation data. Conceptually, these parameters take the role of transport coefficients. A goal of future investigations would be to derive the values of these coefficients from first principles. Furthermore, investigating the effects of external driving such as shear, see [97] for a mode-coupling study of glassy states, would be worthwhile, as would be to relate to the stress correlation function [98].

Describing the short-time behaviour of the van Hove function remains a significant challenge. Improvement of our approximation could potentially be achieved by augmenting the dependence on the weighted density $n_3(\mathbf{r}, t)$ by incorporating further fundamental measure weighted densities. Additionally, the functionals that we apply here can be viewed as a low-order series expansion in powers of the velocity field. Since the velocity field is very large at early times compared to later times, functionals with higher orders of $\mathbf{v}$ might be needed to achieve better agreement in this regime.

## Conflicts of interest

There are no conflicts of interest to declare.

## Acknowledgements

We thank Daniel de las Heras, Roland Roth, Daniel Stopper, and the referees for useful comments. This work is supported by the German Research Foundation (DFG) via Project no. 317849184.

# A    DDFT integration scheme in spherical coordinates

We derive a discrete integration scheme in spherical coordinates as used in our implementation starting from the DDFT equation of motion (23). We set $f_{\text{ext},\alpha} = 0$ and divide by $\gamma$ to obtain

$$\dot{\rho}_\alpha(\mathbf{r},t) = D\nabla^2\rho_\alpha(\mathbf{r},t) + \nabla \cdot \gamma^{-1}\rho_\alpha(\mathbf{r},t)\nabla\frac{\delta F_{\text{exc}}[\{\rho_\alpha\}]}{\delta\rho_\alpha(\mathbf{r},t)}\,. \tag{66}$$

For compactness of notation, we define the excess current profile

$$\mathbf{J}_{\text{exc},\alpha}(\mathbf{r},t) = -\gamma^{-1}\rho_\alpha(\mathbf{r},t)\nabla\frac{\delta F_{\text{exc}}[\{\rho_\alpha\}]}{\delta\rho_\alpha(\mathbf{r},t)}\,, \tag{67}$$

which is a functional of $\rho_\alpha$. Using this, we can write equation (66) as

$$\dot{\rho}_\alpha(\mathbf{r},t) = D\nabla^2\rho_\alpha(\mathbf{r},t) - \nabla \cdot \mathbf{J}_{\text{exc},\alpha}(\mathbf{r},t)\,. \tag{68}$$

In our system, where $\rho_\alpha$ is radially symmetric, the excess current profile can be written as

$$\mathbf{J}_{\text{exc},\alpha}(\mathbf{r},t) = J_{\text{exc},\alpha}(r,t)\hat{\mathbf{e}}_r\,. \tag{69}$$

Equation (68) then simplifies to

$$\dot{\rho}_\alpha(r,t) = D\frac{1}{r^2}\frac{\partial}{\partial r}\left(r^2\frac{\partial}{\partial r}\rho_\alpha(r,t)\right) - \frac{1}{r^2}\frac{\partial}{\partial r}\left(r^2 J_{\text{exc},\alpha}(r,t)\right)\,. \tag{70}$$

In a numerical calculation, $\rho_\alpha(r,t)$ and $J_{\text{exc},\alpha}$ can be represented as arrays of numbers corresponding to equally spaced sampling points of the respective continuous function. We choose some discretisation step $\Delta r \ll \sigma$ in space and some discretisation time $\Delta t \ll \tau$. We can then approximate the density profile as

$$\rho_\alpha(r,t_k) \approx \tilde{\rho}_\alpha(r,t_k) \equiv \sum_{i=0}^{\infty}\rho_{i,k}^\alpha b\left(r - i\Delta r\right)\,, \tag{71}$$

where $i$ is a spatial index, $k$ is a temporal index, and $b(r)$ is a triangle function defined as

$$b(r) = \begin{cases} 1 + r/\Delta r & \text{for } -\Delta r \leq r \leq 0 \\ 1 - r/\Delta r & \text{for } 0 \leq r \leq \Delta r \\ 0 & \text{elsewhere}\,. \end{cases} \tag{72}$$

The same discretisation procedure can be done for the current profile. The spatial and temporal derivatives in (70) can be evaluated using the well-known finite difference formulae to obtain an approximate solution to the partial differential equation (70) for a given initial value for $\rho_\alpha(\mathbf{r},t)$:

$$\rho_{i,k+1}^\alpha = \rho_{i,k}^\alpha - \Delta t\left[\frac{2}{i\Delta r}J_{i,k}^{\text{exc},\alpha} + \frac{J_{i+1,k}^{\text{exc},\alpha} - J_{i-1,k}^{\text{exc},\alpha}}{2\Delta r} + D\left(\frac{\rho_{i+1,k}^\alpha - 2\rho_{i,k}^\alpha + \rho_{i-1,k}^\alpha}{\Delta r^2} + \frac{\rho_{i+1,k}^\alpha - \rho_{i-1,k}^\alpha}{i\Delta r^2}\right)\right]\,. \tag{73}$$

However, using this straightforward approach results in a subtle problem: particle conservation is violated. This can be seen by integrating the difference $\tilde{\rho}_\alpha(r,t_{k+1}) - \tilde{\rho}_\alpha(r,t_k)$ from $r = 0$ to some cutoff $r_{\max}$ and reordering the resulting sum by the spatial index $i$. The result is nonzero and, more importantly, it is not equal to the total particle flux through the surface at $r = r_{\max}$.

This defect can be addressed by re-writing equation (70) with respect to the functions

$$R_\alpha(r,t) \equiv 4\pi r^2 \rho_\alpha(r,t) \quad \text{and} \tag{74}$$

$$J_\alpha^R(r,t) \equiv 4\pi r^2 J_{\text{exc},\alpha}(r,t), \tag{75}$$

which represent the radial density in and current through the whole spherical shell around the origin at $r$. Thereby, we obtain a new partial differential equation

$$\dot{R}_\alpha(r,t) = D\frac{\partial^2}{\partial r^2}R_\alpha(r,t) - D\frac{\partial}{\partial r}\left(\frac{2R_\alpha(r,t)}{r}\right) - \frac{\partial}{\partial r}J_\alpha^R(r,t), \tag{76}$$

which can be discretised using standard finite differences to obtain the iteration scheme

$$R_{i,k+1}^\alpha = R_{i,k}^\alpha + \frac{D\Delta t}{\Delta r}\left[\frac{R_{i+1,k}^\alpha - 2R_{i,k}^\alpha + R_{i-1,k}^\alpha}{\Delta r} - \frac{R_{i+1,k}^\alpha}{(i+1)\Delta r} + \frac{R_{i-1,k}^\alpha}{(i-1)\Delta r} - \frac{J_{i+1,k}^{R,\alpha} - J_{i-1,k}^{R,\alpha}}{2D}\right], \tag{77}$$

where $R_{i,k}^\alpha$ and $J_{i,k}^{R,\alpha}$ are defined as

$$R_{i,k}^\alpha \equiv 4\pi(i\Delta r)^2\rho_{i,k}^\alpha \quad \text{and} \tag{78}$$

$$J_{i,k}^{R,\alpha} \equiv 4\pi(i\Delta r)^2 J_{i,k}^{\text{exc},\alpha}. \tag{79}$$

Using this scheme significantly reduces the deviations in the normalisation of the density profile which we observed in our numerical calculations.

At each time step, the transformation from $\rho_{i,k}^\alpha$ to $R_{i,k}^\alpha$ needs to be inverted in order to calculate $\mathbf{J}_{\text{exc},\alpha}$. The same is necessary if the density profile should be saved to disk in a simulation run. This results in a division by zero for $\rho_{0,k}^\alpha$. To circumvent this issue, we define $\rho^\alpha(0,t)$ as the continuous continuation of $R_\alpha(r,t)/r^2$ to $r = 0$. In practice, we extrapolate from $\rho_{1,k}^\alpha$ and $\rho_{2,k}^\alpha$ logarithmically:

$$\rho_{0,k}^\alpha \approx \exp\left[\left(4\ln\rho_{1,k}^\alpha - \ln\rho_{2,k}^\alpha\right)/3\right]. \tag{80}$$

We note that after the inverted transformation, the density normalisation of $\rho_{0,k}^\alpha$ is not strictly conserved either. However, the shell normalisation of $R_{0,k}^\alpha$ is conserved which prevents the drift of the density normalisation, that occurs in the integration scheme equation (73). The full implementation of this integration scheme is available as open source software [99]. We encourage the interested readers to examine and extend it for their own research.

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
