# Peer review of "Dynamic Decay and Superadiabatic Forces in the van Hove Dynamics of Bulk Hard Sphere Fluids"

_SciPost Physics, doi:SciPost Phys. 12, 133 (2022)_

## Round 1 · Referee Report · Martin Oettel · 2022-2-9

Strengths
1 - complete exposition of a theory for the van Hove functions in the language
of dynamic density functional and power functional theory
2 - full determination by simulation of all contributing forces identified
in this framework ("exact" reference results)
3 - power functional expressions for these forces are not specific to the van Hove
setting but are general for nonequilibrium Brownian systems
4 - link between drag force and long-time self diffusion
Weaknesses
1 - Fits for almost all parameters of the power functional are on the
van Hove data themselves (not from a "general" theory or different systems)
Report
This ms reports an "anatomical study" of the self and distinct van Hove
function for hard spheres at a particular density (rho* = 0.73). Some
comparison to LJ data from previous work is made.
The van Hove functions are viewed as time-dependent density profiles of
the self particle Gs and of the distinct particles Gd. Their time derivative
is a divergence of a current, i.e. proportional to a force density (in BD).
The force density is split into adiabatic and superadiabatic terms according
to the framework of PFT as developed by the group. All terms can be determined
in simulations, thus "exact" results are available to compare with theoretical
approximations.
The theoretical analysis focuses on an in-depth analysis of DDFT results and
their extension using PFT. DDFT gives the exact adiabatic dynamics if the exact
equilibrium functional for the two-species mixture of self and distinct particles
is known. The paper analyses the best currently available functionals for this
mixture and finds deficiencies.
Going beyond DDFT, three types of superadiabatic
forces are taken into account (drag, viscoelastic, structural). These general
types had been identified before and they show distinct signatures in different
settings. They are derived from a certain approximation for the power functional.
A highlight is the derived link of long-time self diffusion constant to the drag
force (viscoelastic and structural forces do not contribute), and the simple
relation for the long-time self diffusion constant to the overall strength of the
drag force.
The types of superadiabatic forces are general and a small velocity expansion
for the corresponding power functional terms has been developed by the group.
However, these terms contain prefactors and memory kernels which were mainly
fitted to the van Hove simulation results themselves, only the viscoelastic term
could be used from an analysis of a different (sheared) system. Overall, the fits
provide a very good medium- to long-time description of Gs and Gd.
I strongly recommend publication of this work. It gives a self-contained theoretical
framework for one of the basic correlation functions in liqiuds which is accessible
both theoretically and experimentally.
Requested changes
1 - I do not fully understand 3.3 and Fig. 5: Do you mean that you calculate
V_ad,a(r) = mu - dF / drho_a(r) (a=s,d)
for different versions of F with rho_a(r) inputted from simulations (for a
particular time)? And then you simulate the equilibrium profile with
this V_ad,a (which should be equal to the rho_a input)?
Perhaps one could write this equation.
Fig 5 is rather small, and the input Gs and Gd curves are hardly distinguishable.
Perhaps use symbols for those?
2 - The quenched functional appears to me rather good in describing the adiabatic
forces but you state it is not reliable. If you use the quenched functional + the
superadiabatic fits, how does it compare to the simulated Gs and Gd?
3 - If you fit the drag amplitude only to the largest times in your simulations,
will DL from (57) get closer to the simulation result? For large times velocities are small and the small velocity expansion should be good.
4 - Can you comment on the relation of the venerable Vineyard approximation for Gd to your framework?
5 - Typo: Medi*n*a-Noyola on p. 2

---

## Round 1 · Referee Report · Anonymous · 2022-2-16

Strengths
1 great introduction to the dynamics of the van Hove function
2 connection to earlier studies of the van Hove function within DDFT
3 connection between the behaviour of hard spheres and a Lennard Jones fluid
Weaknesses
1 Fig. 3 is difficult to read
Report
In this manuscript the authors study the dynamics of the van Hove function
of a hard-sphere fluid. This problem has been studied before with various
theoretical approaches, but the present manuscript makes use of some ideas
in Power Functional Theory (PFT) which is a variational approach to dynamical
systems, out of equilibrium. This additional use of PFT is a clear extension
to earlier studies that employed Dynamical Density Functional Theory (DDFT).
The manuscript gives a detailed and good to read introduction to the DDFT/PFT
approach to the dynamics of the van Hove function and discusses the application
to hard spheres in detail. Moreover, there is an additional insight by
comparing some details of the van Hove functions of hard spheres and that of
a Lennard-Jones fluid.
There are a few minor points that I have picked up while reading (with great
joy, if I may add) through the manuscript.
The authors mention that a shortcoming of DDFT is that it does not pick up
all the details of the out of equilibrium dynamics of a system. To this I
fully agree. A bit later the author mention that even an equilibrium system
shows some dynamics and give the example of the dynamical decay of the van
Hove function. These two statements are somewhat separated in the manuscript,
but if I put them next to each other it seems that either I have missed some
details, by taking these statements out of the context of the manuscript, or
the authors should clarify this point. With the discussion of the force
densities in Fig. 6 it seems that DDFT is missing some contributions even
if the system considered is in equilibrium.
Along a similar line I would like the authors to clarify their discussion
of the long time diffusion in Sec. 2.6. Is it correct that if one were to
apply standard DDFT using the ansatz in Eq.(48) one would obtain Eq.(57)
with C_drag = 0? Is this super-adiabatic contribution to the modified
Einstein relation the answer, or part of the answer, to my previous question?
It would be helpful if the authors could mention this at the end of Sec.
2.6.
One final point is that I find Fig.3 rather difficult to read. The combination
of color and Morse code challenging. Maybe it would help to name the lines
of the plot at t/tau=2 from top to bottom or the other way around, so that the
results are easy to identify.
Once these minor points have taken into account, I can highly recommend the
publication of this manuscript in Sci Post.
Requested changes
1 clarify the role of DDFT and the equilibrium dynamics of the van Hove function
2 clarify the role C_drag in 'standard' DDFT
3 clarify Fig. 3

---

## Round 2 · Author Response

Warnings issued while processing user-supplied markup:

  • Inconsistency: Markdown and reStructuredText syntaxes are mixed. Markdown will be used.
    Add "#coerce:reST" or "#coerce:plain" as the first line of your text to force reStructuredText or no markup.
    You may also contact the helpdesk if the formatting is incorrect and you are unable to edit your text.

Reply to the Anonymous Report 2 on 2022-2-16 (Invited Report)

We thank the Referee for the detailed and positive report. We are delighted that the Referee enjoyed reading our paper and we acknowledge the Referee's constructive and valid points of criticism. We have addressed these points in the revised version of the paper as laid out in the following.

1) The referee demands clarification of the status of the DDFT in describing the dynamics in equilibrium. We agree with the Referee that the DDFT does not give a full description of the nonequilibrium dynamics. The point is to clarify the situation in equilibrium, which indeed is both subtle and important. In the following we first lay out the physics and then describe the changes to the paper.

In the one-body framework, equilibrium implies that the one-body current vanishes and that the one-body density does not change in time. This is the case for the present quiescent bulk fluid. Per construction, DDFT satisfies this static limit, as the corresponding equation of motion reduces to the (gradient of the) Euler-Lagrange equation of DFT, which is formally exact. The power functional theory retains this limit, as all additional contribution, i.e. those of the superadiabatic type, vanish due to the vanishing one-body flow in equilibrium.

The van Hove function as a dynamic two-body correlation function is "hidden from view" in the above one-body picture. Nevertheless, in an equilibrium system non-trivial dynamics occurs on the two-body level of correlations, as is the central topic of the paper. While these correlations are of equilibrium two-body nature, via the dynamical test particle limit, they can analogously be considered to be nonequilibrium one-body density profiles. The mapping from equilibrium two-body functions to nonequilibrium one-body functions is a conceptual trick that requires defining initial condition for one-body profiles that reflect the behaviour of the van Hove function at time zero, i.e. the self part is a delta function and the distinct part is the static g(r), up to a multiplicative constant.

To clarify and emphasize this point, we added the following discussion to the paper at the end of section 3, i.e. after the description of figure 6 (here the numbering is according to the orignal manuscript).

   "Hence the results shown in figure 6 demonstrate explicitly the
    perhaps expected shortcoming of the DDFT that it does not
    capture the full dynamics of the system. Here the van Hove
    function plays a dual role in that it is both an equilibrium
    dynamical correlation function as well as a specific
    nonequilibrium temporal process. The latter is specified by the
    dynamical test particle limit and we recall our description of
    how this approach allows to identify two- with one-body
    correlation functions in section 2.3. As the DDFT does not
    provide the full nonequilibrium dynamics on the one-body
    level, by extension it also fails to describe all forces that
    govern the time evolution of the van Hove function. These
    findings are consistent with the nonequilibrium
    Ornstein-Zernike framework [41,42], where both adiabatic and
    superadiabatic direct correlation functions occur, and only
    the former are generated from the free energy density
    functional. The superadiabatic time direct correlation
    functions are genuine dynamical objects. The present study
    sheds light on this issue in the test particle picture."

Referee: even an equilibrium system shows some dynamics, such as the dynamical decay of the van Hove function.

     if I put these two statements next to each other it seems
     that either I have missed some details. The authors should
     clarify this point.

     With the discussion of the force densities in Fig. 6 it
     seems that DDFT is missing some contributions even if the
     system considered is in equilibrium.

2) Concerning the long time diffusion behaviour, the Referee correctly points out that on the basis of the standard DDFT going through the same chain of arguments gives our result in Eq.(57), but with vanishing drag coefficient, i.e. C_drag=0. Physically, the drag force originates from the friction caused by the interflow of two components, where one of the two component consists of the single "self" particle in the present case. Drag is a genuine flow effect that cannot be captured by the free energy description that underlies the DDFT.

We find the connection from drag to the slowing down of the dynamics to be very striking, as within the bare DDFT it remains a mystery why slow down happens. That drag can be rationalized by a simple power functional [Eq.(37)], that is quadratic in the velocity difference of the two species, is an insight first gained for active particles. As the superadiabatic power functional contributions are intrinsic, i.e. they do not depend on the type of driving but rather on the flow that occurs in the system. Hence drag forces occur also in a passive system where no explicit driving occurs, provided that nonequilibrium is generated by the specific initial condition of the van Hove function at time zero [to match the pair distribution function g(r)].

To alert the reader to this situation, we added the following clarification at the end of Sec.2.6.

  "The decrease of D_L that results from a finite value C_drag > 0
   according to equation (57) is an entirely superadiabatic
   effect. Going through the above derivation on the basis of the
   DDFT alone leads to the trivial results D_L = k_BT/gamma,
   analogous to. formally setting C_drag = 0 in equation
   (57). Hence the power functional ansatz for P_t^drag, see
   equations (37) and (55), links drag (due to interflow of the
   self and distinct components) with the long time self diffusion
   coefficient."

Along a similar line I would like the authors to clarify their discussion of the long time diffusion in Sec. 2.6. Is it correct that if one were to apply standard DDFT using the ansatz in Eq.(48) one would obtain Eq.(57) with C_drag = 0? Is this super-adiabatic contribution to the modified Einstein relation the answer, or part of the answer, to my previous question? It would be helpful if the authors could mention this at the end of Sec. 2.6.

3) We have improved the readability of figure 3. Please see the revised graphics. There are now labels that specify directly to which data set each curve corresponds.

We thank the Referee once more for a stimulating report and for the kind and positive assessment.

Strengths 1 great introduction to the dynamics of the van Hove function 2 connection to earlier studies of the van Hove function within DDFT 3 connection between the behaviour of hard spheres and a Lennard Jones fluid

Weaknesses 1 Fig. 3 is difficult to read

Report In this manuscript the authors study the dynamics of the van Hove function of a hard-sphere fluid. This problem has been studied before with various theoretical approaches, but the present manuscript makes use of some ideas in Power Functional Theory (PFT) which is a variational approach to dynamical systems, out of equilibrium. This additional use of PFT is a clear extension to earlier studies that employed Dynamical Density Functional Theory (DDFT). The manuscript gives a detailed and good to read introduction to the DDFT/PFT approach to the dynamics of the van Hove function and discusses the application to hard spheres in detail. Moreover, there is an additional insight by comparing some details of the van Hove functions of hard spheres and that of a Lennard-Jones fluid.

There are a few minor points that I have picked up while reading (with great joy, if I may add) through the manuscript.

The authors mention that a shortcoming of DDFT is that it does not pick up all the details of the out of equilibrium dynamics of a system. To this I fully agree. A bit later the author mention that even an equilibrium system shows some dynamics and give the example of the dynamical decay of the van Hove function. These two statements are somewhat separated in the manuscript, but if I put them next to each other it seems that either I have missed some details, by taking these statements out of the context of the manuscript, or the authors should clarify this point. With the discussion of the force densities in Fig. 6 it seems that DDFT is missing some contributions even if the system considered is in equilibrium.

Along a similar line I would like the authors to clarify their discussion of the long time diffusion in Sec. 2.6. Is it correct that if one were to apply standard DDFT using the ansatz in Eq.(48) one would obtain Eq.(57) with C_drag = 0? Is this super-adiabatic contribution to the modified Einstein relation the answer, or part of the answer, to my previous question? It would be helpful if the authors could mention this at the end of Sec. 2.6.

One final point is that I find Fig.3 rather difficult to read. The combination of color and Morse code challenging. Maybe it would help to name the lines of the plot at t/tau=2 from top to bottom or the other way around, so that the results are easy to identify.

Once these minor points have taken into account, I can highly recommend the publication of this manuscript in Sci Post.

Requested changes

1 clarify the role of DDFT and the equilibrium dynamics of the van Hove function

2 clarify the role C_drag in 'standard' DDFT

3 clarify Fig. 3

Validity: Top Significance: Top Originality: Top Clarity: Top Formatting: Excellent Grammar: Excellent

Reply to the Report 1 by Martin Oettel on 2022-2-9 (Invited Report)

We thank the Referee for a detailed and constructive report and for the positive assessment. The Referee's points of criticism are valid and helpful. We describe our responses and changes to the manuscript below.

Strengths

1 - complete exposition of a theory for the van Hove functions in the language of dynamic density functional and power functional theory

2 - full determination by simulation of all contributing forces identified in this framework ("exact" reference results)

3 - power functional expressions for these forces are not specific to the van Hove setting but are general for nonequilibrium Brownian systems

4 - link between drag force and long-time self diffusion

Weaknesses

1 - Fits for almost all parameters of the power functional are on the van Hove data themselves (not from a "general" theory or different systems)

Report

This ms reports an "anatomical study" of the self and distinct van Hove function for hard spheres at a particular density (rho* = 0.73). Some comparison to LJ data from previous work is made.

The van Hove functions are viewed as time-dependent density profiles of the self particle Gs and of the distinct particles Gd. Their time derivative is a divergence of a current, i.e. proportional to a force density (in BD). The force density is split into adiabatic and superadiabatic terms according to the framework of PFT as developed by the group. All terms can be determined in simulations, thus "exact" results are available to compare with theoretical approximations.

The theoretical analysis focuses on an in-depth analysis of DDFT results and their extension using PFT. DDFT gives the exact adiabatic dynamics if the exact equilibrium functional for the two-species mixture of self and distinct particles is known. The paper analyses the best currently available functionals for this mixture and finds deficiencies.

Going beyond DDFT, three types of superadiabatic forces are taken into account (drag, viscoelastic, structural). These general types had been identified before and they show distinct signatures in different settings. They are derived from a certain approximation for the power functional. A highlight is the derived link of long-time self diffusion constant to the drag force (viscoelastic and structural forces do not contribute), and the simple relation for the long-time self diffusion constant to the overall strength of the drag force.

The types of superadiabatic forces are general and a small velocity expansion for the corresponding power functional terms has been developed by the group. However, these terms contain prefactors and memory kernels which were mainly fitted to the van Hove simulation results themselves, only the viscoelastic term could be used from an analysis of a different (sheared) system. Overall, the fits provide a very good medium- to long-time description of Gs and Gd.

I strongly recommend publication of this work. It gives a self-contained theoretical framework for one of the basic correlation functions in liquids which is accessible both theoretically and experimentally.

Requested changes

1 - I do not fully understand 3.3 and Fig. 5: Do you mean that you calculate V_ad,a(r) = mu - dF / drho_a(r) (a=s,d) for different versions of F with rho_a(r) inputted from simulations (for a particular time)? And then you simulate the equilibrium profile with this V_ad,a (which should be equal to the rho_a input)? Perhaps one could write this equation. Fig 5 is rather small, and the input Gs and Gd curves are hardly distinguishable. Perhaps use symbols for those?

1) The Referee describes the situation and our method correctly. These calculations are indeed carried out at a fixed time. We agree that stating the Euler-Lagrange equation in the way that the Referee indicates is helpful, as it makes the description of the procedure fully explicit. We hence have added corresponding text in the first paragraph of section 3.3.

"To test whether these functionals are accurate, we calculate the
 adiabatic potential

        V_ad,a (r, t) = mu − dF/drho_a(r,t) (65)

 using each of the three free energy functionals, with the density
 profiles sampled in BD simulation at times t = 0.1, 0.3, 0.6 and
 1.0 tau. Then we run equilibrium BD simulations with an external
 force field fext,a (r, t) = −nabla V_ad,a(r, t) designed to
 cancel out the predicted adiabatic internal force field [65,
 67]."

As concerns Fig.5 (Fig.6 in the revised paper), we are grateful for the useful suggestion to use symbols for the "target" data. We have revised the figure accordingly.

2 - The quenched functional appears to me rather good in describing the adiabatic forces but you state it is not reliable. If you use the quenched functional + the superadiabatic fits, how does it compare to the simulated Gs and Gd?

2) We have chosen the direct simulation approach over the use of the quenched functional in order to avoid error propagation. Hence the adiabatic forces are represented by numerically exact results. This allows us to cleanly separate the modelling of the superadiabatic contributions from the adiabatic problem. The latter problem is surely delicate in its own right, as the test particle setup represents a strongly confined situation. Our strategy hence does not represent an undue criticism of the quenched functional and we agree with the referee that the theory by Stopper et al indeed describes a challenging situation very well. We have clarified that the inaccuracies that the theory exhibits only occur close to the origin. We added the following text.

   "For the quenched functional, the agreement is markedly
    improved, but relevant deviations remain near the origin. As
    we will see below, the forces calculated in this region, via
    the quenched functional, are still not reliable enough to
    facilitate an accurate splitting of the internal forces into
    adiabatic and superadiabatic contributions (see figure 7,
    panel b).

    [...]

    Additionally, we can see in panel b of figure 6 that even the
    quenched approximation for the adiabatic force density shows
    some deviations in the distinct component near the origin.
    These deviations are of similar magnitude as the force itself
    and may thus not be neglected when trying to split the
    internal force field into adiabatic and superadiabatic
    contributions. However, we see that the distinct force density
    component for r>sigma, as well as the self component, are
    accurately reproduced."

As to the suggested comparison, the differences in forces will be those between quenched prediction and reality. The existing figure 5 is already dedicated to performing this demonstration. For further illustration, we have added the quenched prediction for the adiabatic forces to figure 6.

3 - If you fit the drag amplitude only to the largest times in your simulations, will DL from (57) get closer to the simulation result? For large times velocities are small and the small velocity expansion should be good.

3) We have carried out the suggested analysis. The results are presented in the following added text (end of section 3.4).

 "Fitting the drag amplitude only to the largest times results in
  C_drag = 2.4 k_BT tau sigma, from which we obtain D_L=0.36
  sigma^2/tau, improving the match with the simulation
  result. This finding indicates that the differential force field
  contains contributions for early times t < tau which are not
  captured well by our approximation for the drag force."

4 - Can you comment on the relation of the venerable Vineyard approximation for Gd to your framework?

4) The Vineyard approximation is very different from our approach, as we trace the physical origin of the forces that govern the temporal changes of the van Hove function. As we show in the paper, an intricate relationship of different types of effective force fields, whether they act differentially between the two species (self and distinct) or on the total van Hove function. In contrast, the Vineyard approximation expresses the dynamics of the distinct part of the van Hove function as a simple convolution of the static g(r) with the dynamical self part of the van Hove function.

We hence agree with the referee that this relationship and contrast are worth spelling out. We have added an additional plot (figure 5 in the revised paper) that illustrates the results from the Vineyard approximation. Here we work on the basis of the simulation data alone. In the paper we argue that this is the best test of the Vineyard concept in its raw form, as no additional assumptions about the shape of the dynamics of the self van Hove function are involved.

The results demonstrate that while the Vineyard approximation yields temporally decaying distinct correlations, significant numerical deviations occur, as compared to the benchmark of the simulated distinct van Hove function. We have added a corresponding figure 7 and accompanying description in the paper.

The added text reads as follows:

   "One early attempt at a theoretical approximation for the van
    Hove function was presented by Vineyard [3, 91].  He proposed
    the relation G_d(r,t) = (g(r) * G_s(r},t))(r), where the
    operator * indicates a convolution, between the self component
    and the distinct component of the van Hove function.  Hopkins
    et al. found that the Vineyard approximation to be a 'fairly
    good approximation' of the van Hove function [47].  As a
    self-consistency check, we use our simulation data to
    calculate the convolution in equation (64) and compare the
    result to the simulation result for G_d(r,t) (see figure 6).
    We find that the Vineyard approximation significantly
    overestimates the rate of decay of the shell structure of the
    distinct van Hove function, in accordance with previous
    findings [47].  Only the minimum at small distances r is well
    reproduced.  This indicates that, even if we had a perfect
    approximation of the behaviour of the self component of the
    van Hove function, we could not use the Vineyard approximation
    to obtain an equally accurate approximation of the distinct
    van Hove function."

We also added to the outlook a brief statement:

   "We have discussed the accuracy of the Vineyard approximation
    on the basis of our simulation data. Whether the Vineyard
    approximation can form a useful ingredient in the study of
    superadiabatic effects in the van Hove function remains to be
    seen."

5 - Typo: Medina-Noyola on p.2

5) Thanks. Corrected.

Further changes to the paper

We added the following reference:

[96] F. Vogel and M. Fuchs, Stress correlation function and linear response of Brownian particles, Eur. Phys. J. E 43, 70 (2020). https://doi.org/10.1140/epje/i2020-11993-4

Added text (penultimate paragraph of section 4)

"[would be worthwhile], as would be to relate to the stress correlation function [96]"

Modified text (final paragraph of section 3.2.2):

"The positive value of r(Gd (r, t)/ρB − 1) at distances larger than the zero is caused by adhesion of distinct particles around the self particle in the initial condition. It depends on the static structure factor of the fluid in sign and magnitude."

We have also fixed several typos throughout the document.

---

## Round 2 · List of Changes

(Page numbers refer to pages in the new version) - p2: Corrected misspelled name "Medina-Noyola" - p3: Replace Roman numerals with Arabic numerals in the references to later sections - p6: Correct 2 spelling mistakes - p7: Correct 1 spelling mistake - p8: Correct 1 spelling mistake - p9: Correct 1 spelling mistake - p13: Added paragraph "The decrease of D_L that results from a finite value C_drag > 0 according to equation (57) is an entirely superadiabatic effect. Going through the above derivation on the basis of the DDFT alone leads to the trivial results D_L = k_BT/gamma, analogous to. formally setting C_drag = 0 in equation (57). Hence the power functional ansatz for P_t^drag, see equations (37) and (55), links drag (due to interflow of the self and distinct components) with the long time self diffusion coefficient." - p20: - Corrected 1 spelling mistake - Modified last paragraph of section 3.2.2 as follows: "The positive value of r(Gd (r, t)/ρB − 1) at distances larger than the zero is caused by adhesion of distinct particles around the self particle in the initial condition. It depends on the static structure factor of the fluid in sign and magnitude." - Added section reading as follows: "One early attempt at a theoretical approximation for the van Hove function was presented by Vineyard [3, 91]. He proposed the relation G_d(r,t) = (g(r) * G_s(r},t))(r), where the operator * indicates a convolution, between the self component and the distinct component of the van Hove function. Hopkins et al. found that the Vineyard approximation to be a 'fairly good approximation' of the van Hove function [47]. As a self-consistency check, we use our simulation data to calculate the convolution in equation (64) and compare the result to the simulation result for G_d(r,t) (see figure 6). We find that the Vineyard approximation significantly overestimates the rate of decay of the shell structure of the distinct van Hove function, in accordance with previous findings [47]. Only the minimum at small distances r is well reproduced. This indicates that, even if we had a perfect approximation of the behaviour of the self component of the van Hove function, we could not use the Vineyard approximation to obtain an equally accurate approximation of the distinct van Hove function." - p21: - Added figure 5 showing the Vineyard approximation, with the caption "Comparison of the distinct van Hove function Gd(r, t), obtained directly using BD simulation (solid yellow lines) and obtained from simulation data using the Vineyard approximation (dashed blue lines)" - Clarified description of the simulation method used to evaluate the DDFT functionals, as follows: "To test whether these functionals are accurate, we calculate the adiabatic potential

        V_ad,a (r, t) = mu − dF/drho_a(r,t) (65)

 using each of the three free energy functionals, with the density
 profiles sampled in BD simulation at times t = 0.1, 0.3, 0.6 and
 1.0 tau. Then we run equilibrium BD simulations with an external
 force field fext,a (r, t) = −nabla V_ad,a(r, t) designed to
 cancel out the predicted adiabatic internal force field [65,
 67]."
- Modified discussion of the accuracy of the quenched functional as follows:
   "For the quenched functional, the agreement is markedly
    improved, but relevant deviations remain near the origin. As
    we will see below, the forces calculated in this region, via
    the quenched functional, are still not reliable enough to
    facilitate an accurate splitting of the internal forces into
    adiabatic and superadiabatic contributions (see figure 7,
    panel b).

    [...]

    Additionally, we can see in panel b of figure 6 that even the
    quenched approximation for the adiabatic force density shows
    some deviations in the distinct component near the origin.
    These deviations are of similar magnitude as the force itself
    and may thus not be neglected when trying to split the
    internal force field into adiabatic and superadiabatic
    contributions. However, we see that the distinct force density
    component for r>sigma, as well as the self component, are
    accurately reproduced."
  • p23: Figure 7: Added quenched functional approximation for adiabatic force density, and added sentence to the caption: "The quenched functional approximation for the adiabatic force density is shown as a solid green line"
  • p25:
    • Added results for a fit of the drag functional to long-time van Hove function: "Fitting the drag amplitude only to the largest times results in C_drag = 2.4 k_BT tau sigma, from which we obtain D_L=0.36 sigma^2/tau, improving the match with the simulation result. This finding indicates that the differential force field contains contributions for early times t < tau which are not captured well by our approximation for the drag force."
    • Added discussion of the role of DDFT and the nonequilibrium nature of the van Hove function in the dynamic test particle picture: "Hence the results shown in figure 6 demonstrate explicitly the perhaps expected shortcoming of the DDFT that it does not capture the full dynamics of the system. Here the van Hove function plays a dual role in that it is both an equilibrium dynamical correlation function as well as a specific nonequilibrium temporal process. The latter is specified by the dynamical test particle limit and we recall our description of how this approach allows to identify two- with one-body correlation functions in section 2.3. As the DDFT does not provide the full nonequilibrium dynamics on the one-body level, by extension it also fails to describe all forces that govern the time evolution of the van Hove function. These findings are consistent with the nonequilibrium Ornstein-Zernike framework [41,42], where both adiabatic and superadiabatic direct correlation functions occur, and only the former are generated from the free energy density functional. The superadiabatic time direct correlation functions are genuine dynamical objects. The present study sheds light on this issue in the test particle picture."
  • p26:
    • Add short paragraph discussing the Vineyard approximation: "We have discussed the accuracy of the Vineyard approximation on the basis of our simulation data. Whether the Vineyard approximation can form a useful ingredient in the study of superadiabatic effects in the van Hove function remains to be seen."
    • Add reference to stress correlation function: "[would be worthwhile], as would be to relate to the stress correlation function [96]"

---

## Editorial Decision

published